# Signals Without Action: A Value Chain Analysis of Luxembourg's 2021 Flood Disaster

Jeff Da Costa[1], Elizabeth Ebert[3], David Hoffmann[3], Hannah L. Cloke[1,2], Jessica Neumann[1]

[1]Department Geography and Environmental Science, University of Reading, RG6 6AB, Reading, United Kingdom

[2]Department of Meteorology, University of Reading, RG6 6ET, Reading, United Kingdom

[3]Bureau of Meteorology, 3001, Melbourne, Victoria, Australia

*Correspondence to*: Jeff Da Costa (j.dacosta@pgr.reading.ac.uk)

**Abstract** Effective Early Warning Systems are essential for reducing disaster risk, particularly as climate change increases the frequency of extreme events. The July 2021 floods were Luxembourg's most financially costly disaster to date. Although strong early signals were available and forecast products were accessible, these were not consistently translated into timely warnings or coordinated protective measures. While response actions were taken during the event, they occurred too late or at insufficient scale to prevent major impacts. We use a value chain approach to examine how forecast information, institutional responsibilities, and communication processes interacted during the event. Using a structured database questionnaire alongside hydrometeorological data, official documentation, and public communications, the analysis identifies points where early signals did not lead to anticipatory action. The findings show that warning performance was shaped less by technical limitations than by procedural thresholds, institutional fragmentation, and timing mismatches across the chain. A new conceptual model, the Waterdrop Model, is introduced to show how forecast signals can be filtered or delayed within systems not designed to process uncertainty collectively. The results demonstrate that forecasting capacity alone is insufficient. Effective early warning depends on integrated procedures, shared interpretation, and governance arrangements that support timely response under uncertainty.

## 1 Introduction

### 1.1 Early Warning Systems

Effective Early Warning Systems are essential for disaster risk reduction. They identify, assess, and monitor upcoming hazards, allowing people to take action to safeguard communities and livelihoods before a hazard event occurs (Glantz and Pierce, 2023; Kelman and Glantz, 2014; Tupper and Fearnley, 2023). Recognising their significance, the United Nations has set an ambitious target through the Early Warnings for All (EW4All) initiative, to ensure that by 2027, everyone on Earth should be covered by an Early Warning System (WMO, 2022).

As hydrometeorological hazards become more frequent and intense, global efforts to expand and improve early warning capabilities have gained renewed urgency (Tupper and Fearnley, 2023; WMO, 2022). Early Warning Systems have therefore become central to disaster risk management (UNDRR, 2015), yet their performance remains inconsistent, even in well-resourced settings (Alfieri et al., 2012).

Early Warning Systems for hydrometeorological hazards consist of interconnected components, including weather and hydrological forecasting, communication technologies and behavioural science (WMO, 2024a). Improving and implementing effective Early Warning Systems requires a holistic, interdisciplinary perspective that

recognises the complex interactions between science, technology, and decision-making (Hermans et al., 2022; Oliver-Smith, 2018).

There is no universally agreed definition of an Early Warning System, as disciplinary and institutional perspectives vary (Kelman and Glantz, 2014). The United Nations Office for Disaster Risk Reduction (UNDRR) defines Early Warning Systems as integrated systems composed of four key elements: risk knowledge, monitoring and warning services, dissemination and communication, and response capability. Such systems aim to enable individuals, communities, and institutions to act in time to reduce disaster risk (UNDRR, 2015; WMO, 2022).

Evaluating the effectiveness of ~~warning system~~Early Warning Systems remains a recognised challenge (Basher, 2006; Coughlan de Perez et al., 2022). While limitations such as institutional fragmentation, interpretive constraints, and procedural rigidity have been widely documented, these issues are often overshadowed by discussions of forecast accuracy or alert delivery (Alcántara-Ayala and Oliver-Smith, 2019; Mileti and Sorensen, 1990). While forecast accuracy and alert dissemination remain important elements of early warning performance, recent work highlights the need to understand how institutional structures, procedures and interpretation processes influence whether available information leads to timely action (Busker et al., 2025; Coughlan de Perez et al., 2022; Diederichs et al., 2023). Each disaster unfolds within a specific context and understanding these conditions is essential for analysing and evaluating ~~warning systems~~Early Warning Systems on a case-by-case basis (Oliver-Smith, 2018).

**1.2 From Forecasts to Action: A Value Chain Approach**

We apply a value chain approach to examine how Early Warning Systems function in practice. The Value Chain Project builds on the World Meteorological Organization (WMO) World Weather Research Programme (WWRP) High Impact Weather (HIWeather) initiative by conceptualising Early Warning Systems as information value chains (Ebert et al., 2023; Hoffmann et al., 2023; WMO, 2024b). The framework aims to improve decision-making by ensuring that each stage of the chain adds value and supports consistent interpretation across institutional actors (WMO, 2024b).

The value chain approach shifts focus from technical accuracy alone to the entire process by which forecasts are interpreted, communicated, and acted upon. This includes the institutional decisions that shape how warning information is transmitted, prioritised or delayed across different actors. The concept of "valleys of death" separating peaks of disciplinary expertise was introduced by Golding (2022) to highlight communication breakdowns across scientific domains. This framing was later expanded by the Value Chain Project, particularly by Hoffmann et al. (2023), who developed a full value chain model that incorporates feedback loops, iterative co-production and institutional decision pathways (Figure 1).

In Luxembourg, early warning and emergency management are organised within a centralised national governance system, with no intermediate regional tier between national authorities and municipalities. Forecasting, warning issuance, emergency planning and crisis coordination are assigned to distinct national authorities. The following sections introduce the national

and transboundary context of the July 2021 floods, while Section 3 provides a detailed description of institutional roles,
responsibilities and activation protocols.

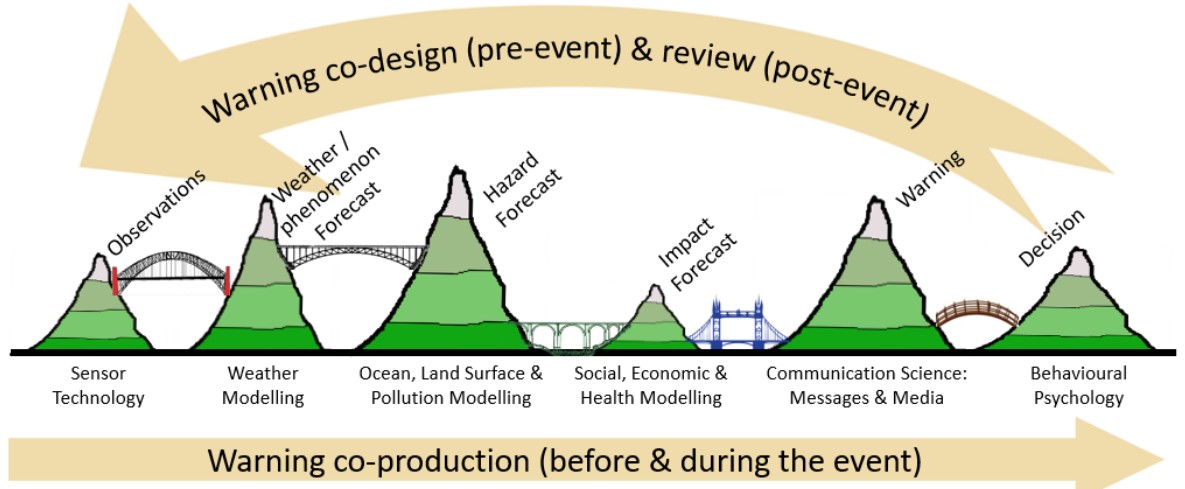


**Figure 1 The warning chain as five "valleys of death" separating peaks of disciplinary expertise,** showing the capabilities and outputs
(mountains) and information exchanges (bridges) linking the capabilities and their associated communities (Tan et al., 2022). Before and
during an actual severe event, the flow of information is predominantly downstream, while for post-event assessments, implementation of
improvements, and creation of new services the chain becomes a feedback loop. Figure originally published in Hoffmann et al. (2023) and
used here with co-author permission.

## 80 1.3 Transboundary Risk and Governance in Luxembourg

Luxembourg lies almost entirely within the Moselle sub-catchment of the Rhine basin (European Commission, 2021). Its
eastern border follows the Moselle, Sauer, and Our rivers. As shown in Figure 2, most of the country lies within a broader
transboundary catchment that connects Luxembourg with Germany, France, and Belgium. Along most of its eastern border,
Luxembourg and Germany jointly administer sections of the Moselle and Sauer and Our rivers through condominium
arrangements (see Box 1). These arrangements assign shared legal responsibility to both countries and do not establish a
fixed national boundary along the rivers (Moselle Convention States, 1956; Our-Sauer-Moselle, 1984; Zaiotti, 2011).
Although these agreements apply only to specific river sections, they highlight a broader reality in which physical risk is
shared across borders, but mandates for managing that risk remain nationally defined (European Commission, 2021).
National authorities remain responsible for issuing forecasts, setting alert thresholds and activating emergency plans within
their own jurisdictions. Cross-border coordination depends on established protocols, but operational decisions are still taken
within national systems (Becker et al., 2018; Schanze, 2009).
Luxembourg is highly integrated with its neighbours. Roughly 47 percent of the workforce commutes daily from
neighbouring countries and over 170 nationalities reside within its borders (STATEC, 2022). Public services operate in
multiple languages, including Luxembourgish, French, and German. While people, services, and information flow fluidly
across borders, responsibility for warning and emergency coordination remains limited to national authorities.

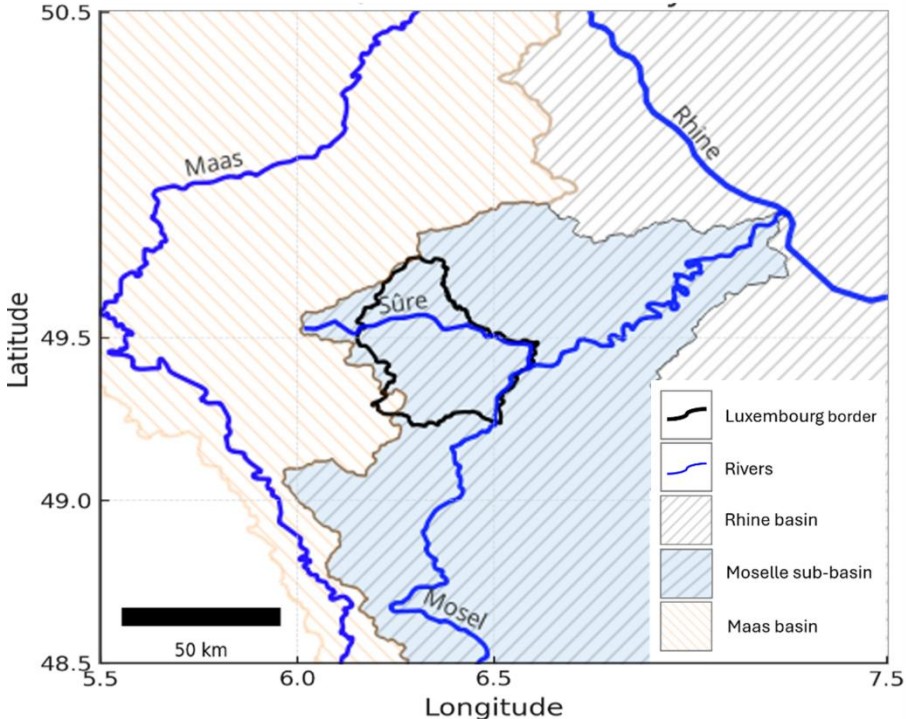

**Figure 2 Luxembourg's position** **the Moselle and Maas catchments (sub-basin of the Rhine basin)**~~within the Rhine basin.~~ The ~~national~~ Luxembourg border (thick black line) outlines ~~Luxembourg~~the national border, which lies almost entirely within the Moselle sub-catchment (blue dashed), itself part of the larger Rhine basin (grey dashed). A small portion in the southwest lies within the Meuse basin (orange dashed). The eastern border follows the Moselle, Sauer, and Our rivers (blue lines), parts of which are governed as international condominiums.

In July 2021, the meteorological conditions that led to flooding developed across the region. While neighbouring countries
experienced similar rainfall and catchment conditions, the warnings issued and decisions taken varied (Busker et al., 2025;
Grimaldi et al., 2023). Whether a hazard event leads to disaster depends not only on the physical event, but on how risk is
interpreted and managed within institutional and social systems. Disasters occur when hazards interact with conditions of
vulnerability, exposure, and governance.~~, rather being a direct outcome of the hazard itself~~ (Ball, 1975; Gould et al., 2016).
Luxembourg provides a relevant case as its location, demographic profile, and degree of cross-border integration make it an
important setting to examine how nationally organised warning and response systems operate in a transboundary context. It
shows that institutional responsibilities influence responses to shared risks. We examine how forecast information ~~was~~were
interpreted and acted upon within this transboundary environment and how institutional structures shaped the management
of the 2021 flood event.

---

**Box 1. River Condominiums** Parts of the Moselle, Sauer and Our rivers form Luxembourg's eastern border with Germany. In these sections, the rivers are governed as condominiums, legal arrangements that grant joint sovereignty to both countries over the entire waterbody. This arrangement originates from Article 27 of the 1816 Treaty of Aachen, which established joint sovereignty over rivers forming the state boundary and later reaffirmed in bilateral treaties in 1984. While cooperation exists on navigation and infrastructure, emergency and warning responsibilities remain defined at the national level even in areas where physical geography is shared but operational governance is not (Moselle Convention States, 1956; Our-Sauer-Moselle, 1984; Treaty of Aachen, 1816; Zaiotti, 2011)

---

### 1.4 The July 2021 European Flood Disaster

In July 2021, extreme rainfall and widespread flooding tested early warning and emergency systems across western Europe. Between 12-15 July, heavy rainfall, saturated soils, and a slow-moving low-pressure system triggered devastating floods in Germany, Belgium, Luxembourg, France and the Netherlands (EUMETSAT, 2021). In Germany alone, the floods caused over 180 fatalities and an estimated €32 billion in losses (Rhein and Kreibich, 2024; Zander et al., 2023). In Luxembourg, the event was the costliest on record, with damages exceeding €145 million and more than 6,500 homes inundated (ACA, 2021). In Luxembourg, the July 2021 floods were formally declared a 'natural disaster', reflecting the scale of impacts relative to national coping capacity rather than absolute losses. While the event was smaller in scale than the catastrophic flooding experienced in parts of Germany, it exceeded available response and recovery capacities in Luxembourg and constituted the most damaging flood event on record nationally. Luxembourg's position within a dense river network contributes to recurrent flood exposure, particularly in low-lying valleys and urbanised catchments.Luxembourg's position within a dense river network contributes to frequent flood exposure, especially in low-lying valleys and urbanised catchments. Historically, major floods occurred in winter, driven by snowmelt and seasonal rainfall, with notable events in 1983, 1993, 1995, 2003, and 2011 (ACA, 2021; AGE, 2021b). These events, though limited in number, have raised concern over a possible shift in seasonal flood patterns. Recent studies suggest that off-season flood risk may be increasing in the region (Ludwig et al., 2023). On 14 July 2021, the Godbrange weather station recorded 105.8 l/m² of rainfall in 24 hours, the highest national daily rainfall total on record.

Although forecasts were available, warnings did not reach higher levelcolour-coded alert levels until shortly before impacts began to unfold. Challenges in communication, including a warning notification via the GouvAlert mobile notification system (Gouvalert) that was not delivered and delays in institutional coordination, contributed to ambiguity regarding responsibilities and the actions expected of different actors. appropriate actions. These factors, combined with limited preparedness across agencies, revealed underlying structural constraints in Luxembourg's Early Warning System (Szönyi et al., 2022).

Germany and Belgium have received substantial scholarly attention (Lietaer et al., 2024; Ludwig et al., 2023; Mohr et al.,
2023; Rhein and Kreibich, 2024; Thieken et al., 2023), but Luxembourg's experience remains comparatively
underexamined. Broader European studies have analysed forecast and warning performance, most notably (Busker et al.,
2025), who provide a synthesis across countries. In these accounts, Luxembourg is only briefly addressed.

**1.5 Learning from the 2021 Flood in Luxembourg**

Using a value chain approach, we reconstruct how forecasts and information was interpreted and shared across agencies and
institutional levels. The analysis traces communication and decision points across the ~~warning system~~Early Warning System
to examine how information moved and what institutional processes shaped the response (Busker et al., 2025; Hagenlocher
et al., 2023). This includes exchanges between national meteorological services, water management authorities, emergency
coordination bodies, and local responders.
To explore how institutional structures may have influenced the timing of ~~response~~ action during the event, we present the
Waterdrop Model, a conceptual model that illustrates how forecast signals interact with organisational constraints and
procedural ~~institutional~~ thresholds for decision-making. The model was developed during post-event reflection and
synthesizes patterns observed in the Luxembourg case and comparable events. It is revisited in section 6.
While the findings are specific to Luxembourg, they reflect broader challenges in countries where early warning depends on
multi-level institutional coordination. This analysis helps clarify how governance structures, communication dynamics and
procedural thresholds shape the performance of ~~warning systems~~Early Warning Systems and their capacity to support timely,
protective action.

# 2. Methods

**2.1 The Value Chain ~~Framework~~ Approach and Database Questionnaire ~~Tool~~**

A central element of the value chain approach is a database questionnaire designed to evaluate Early Warning Systems
performance. It builds on the WMO WWR HIWeather Value Chain Project, which conceptualise Early Warning Systems as
information chains that extend from forecast generation to community-level protective action, including measures taken by
individuals, communities, and institutions (Ebert et al., 2023; Hoffmann et al., 2023; WMO, 2024b). The questionnaire is
maintained by the University College London (UCL) Warning Research Centre (Ebert et al., 2024; UCL, 2025).
The database questionnaire combines quantitative and qualitative inputs to assess how weather information moves through
the warning chain, including bulletins, official statements and institutional actions. It is structured around a sequence of
value chain stages and was designed to capture technical, institutional and communication-related factors (Ebert et al., 2024;
Hoffmann et al., 2023). The approach differs from traditional forecast evaluation methods by focusing on how warnings are
understood, interpreted and ~~acted upon~~translated into action by different actors across the chain.
We completed the standard version of the ~~database~~ questionnaire retrospectively using available public records, institutional
documentation and supplementary datasets. The completed questionnaire will be archived with the UCL Warning Database[1]
and made available upon publication. Figure 3 provides a schematic overview of this methodological structure.

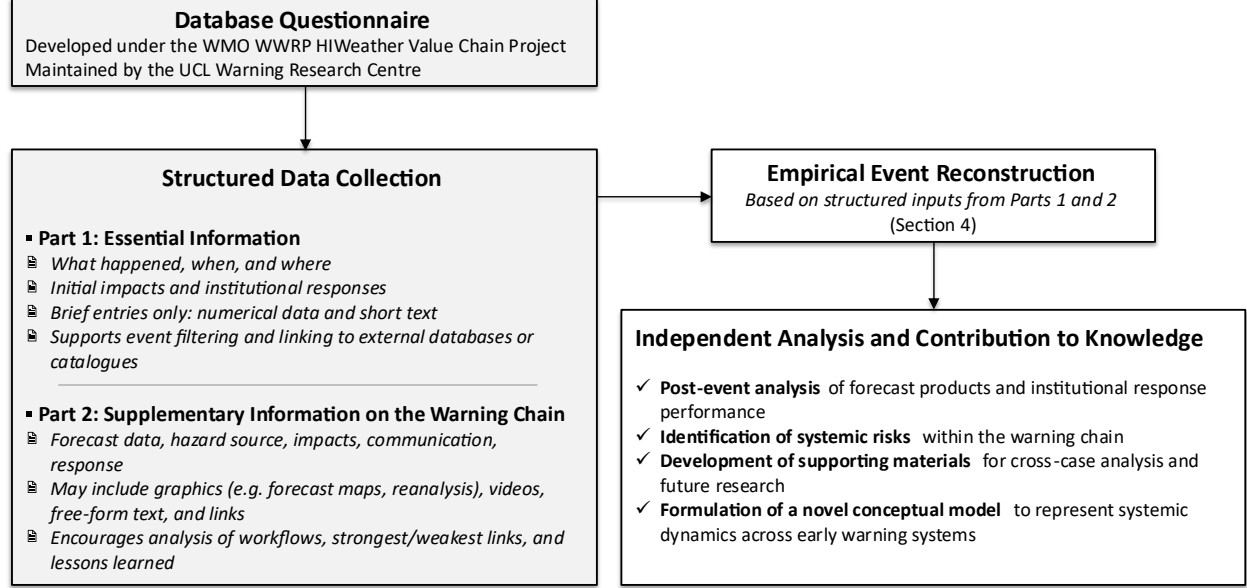


**Figure 3 Schematic representation of the methodological structure used.** The structure of the database questionnaire (Part 1: Essential
Information; Part 2: Supplementary Information on the Warning Chain) is adapted from Hoffmann et al. (2023). The original questionnaire
also includes Part 3, a subjective effectiveness rating, which was not used in this study. These inputs also informed the Waterdrop Model
presented in Section 6.
**2.2 Applying the Value Chain Approach to the 2021 Flood in Luxembourg**
We applied the database questionnaire to the July 2021 floods in Luxembourg to reconstruct how forecasts were generated,
interpreted and communicated, and how decisions were made within national institutions. The analysis focuses on what
information was available, how it was interpreted and how it shaped the activation of protective measures and emergency
plans. In addition to the questionnaire, we drew on multiple forensic analysis frameworks to examine how decisions were
made, including Forensic Investigations of Disasters (FORIN) (Alcántara-Ayala and Oliver-Smith, 2016) and the Post-Event
Review Capability (PERC) (Szönyi et al., 2022). These frameworks aim to identify underlying risk drivers and institutional
barriers to effective ~~response~~action.
We used a structured timeline-based approach to organise institutional messages, ~~alert levels~~colour-coded alert levels and
decision points. This included bulletin releases, agency communications and reported emergency actions. Forecast and

---

[1] UCL. (2025). UCL Warning Database. Warning Research Centre, University College London.
https://www.ucl.ac.uk/sts/warning-research-centre/ucl-warning-database

reanalysis data were sourced from the ECMWF Severe Weather Catalogue (Magnusson, 2019), ERA5 reanalysis (Hersbach
et al., 2020), and the European Severe Storms Laboratory (ESSL) (www.essl.org).
Operational mapping from the Copernicus Emergency Management Service (CEMS) (https://emergency.copernicus.eu) and
event reporting from the international disaster database (EM-DAT) (www.emdat.be) supplemented the analysis. We also
used grey literature, press releases, social media and institutional archives to reconstruct public messaging, institutional
coordination and informal communication dynamics. Information was reviewed in three working languages
(Luxembourgish, French, German), and findings were triangulated across sources. Where available, supplementary data
were accessed through institutional partnerships or publicly released repositories.

## 3. Institutional and Legal Framework for Disaster Management in Luxembourg


**3.1 Institutional Roles and Responsibilities**

The institutional framework for weather and flood forecasting and emergency response in Luxembourg is centralised at the
national level but implemented through coordination between ministries, public agencies and municipalities.
The Ministry of Home Affairs is responsible for emergency planning and collaborates with~~supervises~~ the High
Commissioner for National Protection (*Haut-Commissariat à la Protection Nationale*, HCPN), the central crisis coordination
body. The HCPN, established in 2016 under the HCPN Law, ~~which~~ leads preparedness and interministerial coordination
under the Prime Minister (HCPN Law, 2016).
The Ministry of the Environment, Climate and Sustainable Development manages water resources and oversees flood
preparedness through the Water Management Administration (*Administration de la Gestion de l'Eau,* AGE). AGE conducts
hydrological monitoring, issu~~ing~~es flood forecasts and warnings, and ~~maintaining~~ maintains the national Flood Forecasting
Service (*Service de Prevision des Crues,* SPC) (HCPN Law, 2016).
The Grand Ducal Fire and Rescue Corps (*Corps Grand-Ducal d'Incendie et de Secours,* CGDIS) is Luxembourg's unified
emergency service agency. Created by the *loi du 27 mars 2018 portant organisation de la sécurité civile* (Law of 27 March
2018 on the Organisation of Civil Security), it merged local fire brigades, emergency medical services, and civil protection
units into a single national structure. CGDIS operates within a multi-hazard civil protection framework, with responsibility
for operational response to meteorological, hydrological and other civil protection emergencies in Luxembourg. ~~CGDIS~~
~~leads operational response during severe weather and flooding and with both municipalities and national coordination bodies~~
(CGDIS Law, 2018). Article 69 of this law also mandates a *Plan N~~n~~ational d'O~~o~~rganisation des S~~s~~ecours* (National
Organisation of Emergency Services Plan, PNOS), which sets national coverage objectives, defines the operational
organisation of rescue services~~,~~ and establishes performance evaluation mechanisms. The PNOS was approved and signed in
October 2021 and had not yet been implemented during the July 2021 flood event. In July 2021, operational response to
floods and severe weather was carried out under the structures established by the CGDIS law and the applicable *Plans*
*d'intervention d'urgence* (*Emergency Intervention Plans*), including the *PIU Inondations* (Flood Emergency Intervention
Plan) and the *PIU Intempéries* (Severe Weather Emergency Intervention Plan).
MeteoLux is the sole national authority for issuing meteorological warnings and forecasts. It operates under the Ministry of
Mobility and Public Works and is part of the Air Navigation Administration (*Administration de la navigation aérienne*),
based at Luxembourg-Findel Airport. All national warning thresholds are based on data from its single official observation
station at Findel. MeteoLux uses a four-colour coded alert level scale (Table 2). While it issues public forecasts and
warnings, it cannot independently activate emergency plans or emergency alert systems. Only alerts issued by MeteoLux are
considered valid for national decision-making. Institutional Meteorological forecasts and warnings issued by MeteoLux are
recognised as the official basis for decision-making, while Crisis Unit activation are determined by the HCPN and the Prime
Ministerthresholds and any Crisis Unit activation must be decided by the HCPN and the Prime Minister (HCPN Law, 2016;
Ministry of State et al., 2015).
AGE monitors river levels through a network of over 30 hydrometric stations and issues flood forecasts and warnings via
www.inondations.lu. Flood warnings are also displayed on www.meteolux.lu alongside meteorological warnings. The Flood
Forecasting Service (*Service de prévision des crues*, SPC), chaired by AGE, applies a three-level vigilance scale (Table 3)
linked to defined update frequencies and bulletin issuance. Under the Flood Emergency Intervention Plan, SPC also advises
the HCPN when procedural hydrological thresholds for institutional activation are reached.
The Technical Agricultural Services Administration (*Administration des services techniques de l'agriculture*, ASTA)
operates a network of more than 35 meteorological stations used for agricultural and environmental monitoring
(www.agrimeteo.lu/Agrarmeteorologie). These stations are not integrated into the official warning systemEarly Warning
System and their data are excluded from formal alert protocols. National decisions rely exclusively on MeteoLux forecasts
(Ministry of State et al., 2015)
The HCPN manages infocrise.lu , Luxembourg's national crisis information portal, which provides official emergency plans,
institutional updates, and public guidance. Official alerts are disseminated via GouvAlert, the national mobile notification
system in place during 2021, following activation by the competent authorities..[2]
*Table 2* presents an overview of the institutions responsible for issuing, interpreting, and implementing warnings in
Luxembourg's disaster risk system.

| Actor | Role | Key Responsibilities |
|---|---|---|
| **Ministry of Home Affairs** | National oversight | Leads disaster risk strategy, supervises HCPN, and coordinates inter-agency emergency response. Reports to parliament. |
| **Ministry for the** | Sectoral coordination | Oversees water resource management and municipal |

---

[2] GouvAlert was replaced by LU-Alert (https://lu-alert.lu/en) in 2024, Luxembourg's current national warning systemEarly Warning System. All analysis here refers to the alerting framework in place during the July 2021 flood event.

| Environment | | flood preparedness; chairs AGE. |
|---|---|---|
| **High Commissioner for National Protection (HCPN)** | National crisis coordination | Maintains emergency plans, oversees crisis evaluation, requests Crisis Unit activation. Manages www.infocrise.lu |
| **Prime Minister** | Executive leadership | Authorises Crisis Unit activation and leads national-level coordination during major crises. |
| **MeteoLux** | Meteorological authority | Issues weather warnings via a four-colour scale via www.meteolux.lu. Uses a single official station (Findel) for national alert thresholds. Cannot activate crisis measures independently. |
| **AGE (Administration de la gestion de l'eau)** | Flood forecasting | Manages flood forecasts and river monitoring. Chairs the SPC. Publishes flood warnings on inondations.lu (also displayed on meteolux.lu) and advises HCPN under the Flood Emergency Intervention plan. |
| **CGDIS (Grand-Ducal Fire and Rescue Corps)** | Emergency services | Leads operational response, evacuation, and public safety during extreme weather and floods. |
| **ASTA (Administration des Services Techniques de l'Agriculture)** | Agrometeorological monitoring | Operates more than 35 weather stations for agriculture. Not integrated into official warning protocols; issues alerts via www.agrimeteo.lu |
| **Municipalities** | Local responders | Implement local flood protection measures and coordinate community-level actions. |
| **Crisis Unit** | Multi-agency coordination | Activated by the Prime Minister. Coordinates strategic response involving HCPN, MeteoLux, AGE, CGDIS, and other bodies. |
| **www.inondations.lu** | Public flood alert platform | Disseminates flood alerts, bulletins, and hydrological information to the public. |
| **www.infocrise.lu** | Government crisis information portal | Provides background on emergency protocols and institutional roles. Not used for real-time alerts. |
| **www.meteolux.lu** | Public weather alert platform | Disseminates official weather warnings issued by MeteoLux and displays flood warnings mirrored from |

| | | inondations.lu. |
|---|---|---|


**Table 1 Roles and responsibilities of national and local actors in Luxembourg's disaster management system.**

### 3.2 Emergency Planning and Activation Protocols

Luxembourg's emergency coordination system for severe weather and floods is defined by emergency intervention plans,
adopted by decree in 2015 (severe weather) and 2019 (floods). These plans set out colour-coded alert levels, institutional
roles and activation procedures (HCPN, 2019; Ministry of State et al., 2015). Both plans use a four-phase colour-coded
warning structure as summarised in *Table 3*.

### 3.2.1 Severe Weather Emergency Intervention Plan

MeteoLux determines warning levels based on procedural rainfall thresholds and duration-intensity curves (HCPN, 2015). It
issues public warnings, but these do not automatically trigger activation of emergency responseplans. Once a red alert level
is issued, an inter-institutional evaluation unit, chaired by MeteoLux, assesses the situation. The HCPN is informed and
determines whether the Crisis Unit should be activated. That decision rests with the Prime Minister and is based on
institutional review rather than forecast level alone (HCPN, 2019).

### 3.2.2 Flood Emergency Intervention Plan

Flood alerts are issued by the SPC, chaired by AGE, based on procedural hydrological thresholds and real-time river data.
Warnings are published through inondations.lu and mirrored on meteolux.lu (AGE, 2021d). These bulletins are shared with
CGDIS, municipalities and the HCPN through institutional channels.
In the red alert phase, AGE must notify the HCPN, which evaluates whether national coordination is needed. As with the
meteorological plan, activation of the Crisis Unit activation is not automatic. It is authorised only when the Prime Minister
concludes that multi-agency coordination is required, typically for complex or cross-border events (HCPN, 2019). Once
activated, the Crisis Unit coordinates national emergency response actions, including evacuation, emergency logistics, and
communication. It includes representatives from HCPN, MeteoLux, AGE, CGDIS, Police, the Army, and other ministries
depending on the scenario (Ministère de l'Intérieur and HCPN, 2021a).

**Table 2 Alert thresholds for rainfall and flood events** (adapted from HPCN, 2019; Ministère d'État et al., 2015; Ministry of Home
Affairs, 2021). Official documentation does not explicitly specify whether thresholds are defined using forecasted or observed data. In
practice during July 2021, rainfall alerts issued by MeteoLux were forecast-based, while flood alerts issued by AGE relied on observed
river levels. Terminology reflects the institutional configuration and official wording in use during July 2021. Subsequent changes
introduced after 2024 are outside the scope of this analysis. Documentation does not explicitly define whether thresholds are based on
forecasted or observed data. In practice, rainfall alerts from MeteoLux are forecast-based, while flood alerts from AGE rely on observed
river levels.

| Emergency Intervention Plan | Alerts | Description | Thresholds set by Emergency intervention plans. |
|---|---|---|---|
| Severe Weather Emergency Plan (for rainfall only) | Green | No danger | NA |
| | Yellow | Potential Danger | NA |

| | | | |
|---|---|---|---|
| | Orange | Danger | 31-45 mm in 6 hours or 51-80 mm in 24 hours |
| | Red | Extreme Danger | More than 45 mm in 6 hours or 80 mm in 24 hours |
| Flood Emergency Plan (Excluding Flash Floods) | Green | No flood risk (normal phase) | NA |
| | Yellow | Potential flood risk (vigilance phase) | Triggered by meteorological conditions, whether observed or forecasted, indicating a potential rise in water levels |
| | Orange | Minor flood risk (pre-alert phase) | Initiated when river levels approach pre-alert levels within 24 hours. |
| | Red | Major flood risk (Alert phase) | Triggered when river levels reach or exceed alert levels. |


## 4. Reconstruction of the Flood in Luxembourg

**4.1 Antecedent Conditions and Rainfall Evolution**

In the months preceding July 2021, Luxembourg experienced frequent precipitation, leading to saturated soils and an elevated risk of surface runoff across much of the country's river basins (EUMETSAT, 2021; Ludwig et al., 2023; Tradowsky et al., 2023). At the same time, sea surface temperatures over the Baltic Sea were more than 8°C above average, increasing atmospheric moisture availability (Lang and Poschlod, 2024). This warm and humid air mass contributed to greater atmospheric instability in the region and conditions became increasingly favourable for extreme precipitation (Mohr et al., 2023).

The critical rainfall event was associated with low-pressure system Bernd, which became quasi-stationary over western Europe due to a blocking anticyclone positioned to the northeast (Mohr et al., 2023). Between 13 and 15 July, regional totals ranged from 100 to 200 mm. On 14 July, the Godbrange station in central Luxembourg (approximately 12 km east-northeast of the Findel station) recorded 105.8 mm in 24 h, the highest national daily total on record (MeteoLux, 2021).

The volume and persistence of rainfall triggered widespread surface runoff and fluvial flooding. Ensemble forecasts began signalling the potential for high rainfall from 7 July onwards, with observed and proxy totals later confirming extreme precipitation across Luxembourg (Figure 4).

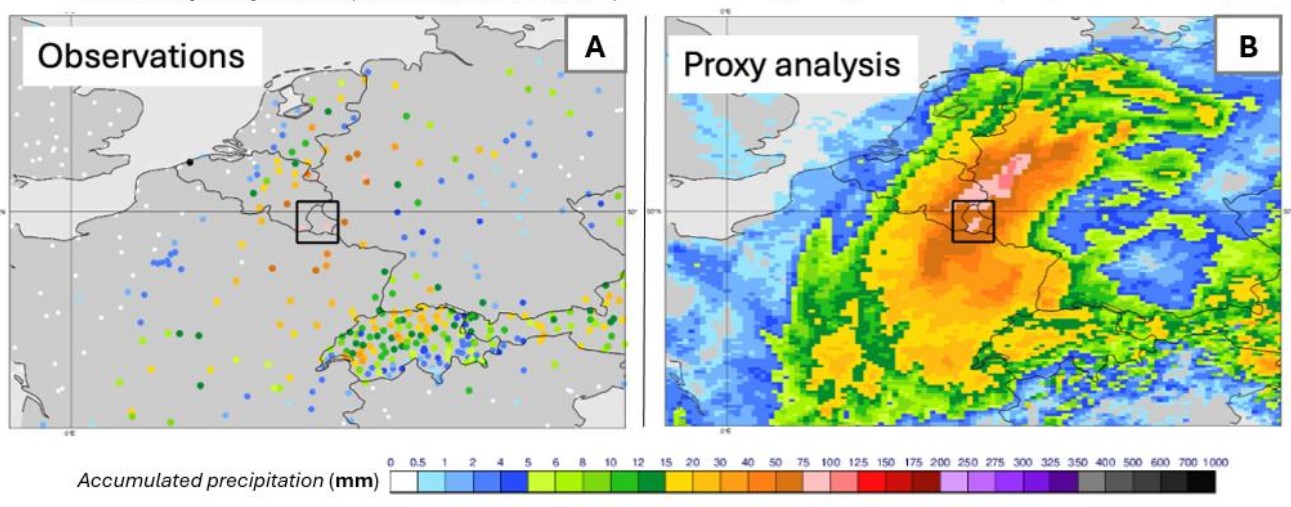

**Observed precipitation** (station measurements)

**Proxy precipitation analysis** (ECMWF-based)

*Accumulated precipitation* (**mm**)

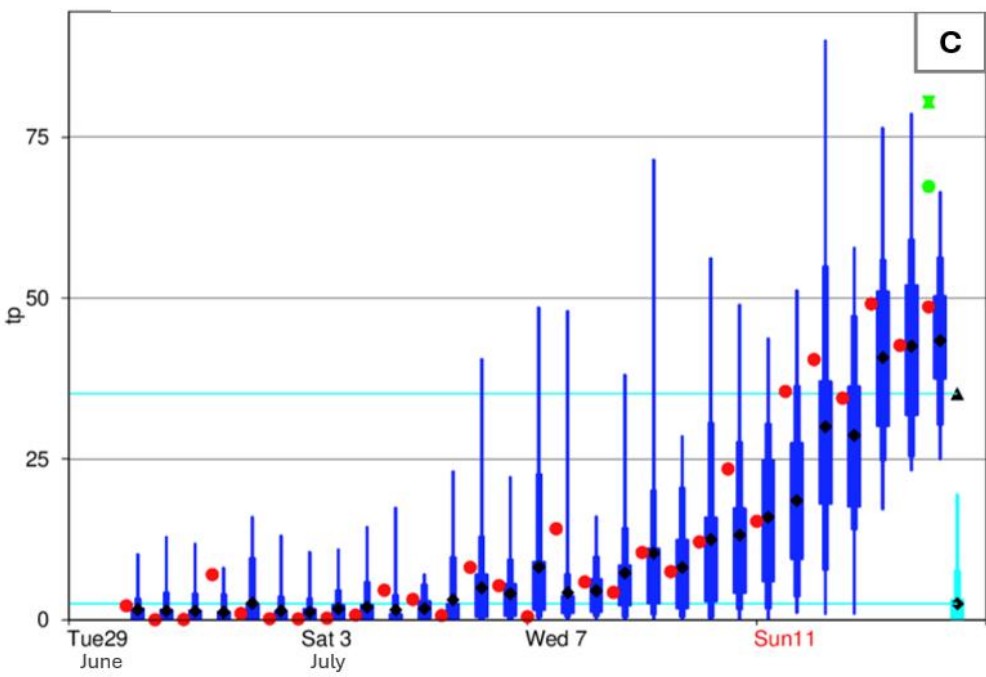

**Figure 4** Observed, proxy and ECMWF ensemble forecast precipitation associated with the 14–15 July 2021 flood event in Luxembourg. (A) Observed precipitation from station measurements for 14–15 July 2021, aggregated over a 1° × 1° grid box centred on 49.75° N, 6° E (Luxembourg). (B) Proxy precipitation analysis for the same period and spatial domain, used where direct observations are spatially or temporally limited. In both panels, colours indicate accumulated precipitation (mm). (C) ECMWF ensemble forecast precipitation evolution over the same 1° × 1° grid box. Blue box-and-whisker plots represent the distribution of IFS ensemble forecast members (IFS-ENS) for each forecast date, red dots indicate the deterministic control forecast, and cyan box-and-whisker plots show the corresponding IFS model climate. Black triangle denotes the maximum value of the model climate. Green hourglass symbol represents the mean of station observations within the analysis box, while the green dot indicates the proxy precipitation totals for 14–15 July. Turquoise horizontal lines denote fixed reference thresholds derived from the model climate. From 7 July onward, forecast spread increases markedly, with some ensemble members exceeding 50 mm, consistent with the high precipitation totals observed in panels A and B.

288

**4.2 Flood Onset and Impacts**

Flood onset began late on 13 July, with sustained rainfall intensifying overnight into 14 July (Mohr et al., 2023). Water levels rose across the country (Douinot et al., 2022). The SPC issued a yellow vigilance alert at 14:30 on 13 July, upgraded to orange by midday on 14 July and to red at 17:15 the same day (AGE, 2021a). At the time of the yellow level alert on 13 July, river levels were already increasing across several catchments. Flooding began during the early hours of 14 July as rainfall intensified and runoff accumulated. By the time the red level alert was issued in the late afternoon of 14 July, flooding was already affecting multiple river systems, with water levels continuing to rise and peak conditions extending into 15 July. Rainfall accumulations in some basins approached or exceeded 100-year return periods, and ~~institutional~~ procedural thresholds for red-alert activation were surpassed at multiple sites (AGE, 2021a; Mohr et al., 2023).

Hydrologically, the event was marked by multi-day discharge exceedances with prolonged peaks in several catchments. In Ettelbruck, water levels remained above warning thresholds for over 30 hours. Most catchments in central and northern Luxembourg experienced prolonged peaks, while the Moselle showed more modest response due to its engineered channel structure (Douinot et al., 2022). Despite occurring in midsummer, the event's discharge profile resembled winter flooding, with high antecedent flow, prolonged flood persistence, and strong basin connectivity (Ludwig et al., 2023).

River levels began receding on 15 July. Emergency damage assessments were initiated the same day by CGDIS and AGE, in coordination with municipal authorities. Clean-up and infrastructure recovery efforts extended through the weekend of 17-18 July (CGDIS, 2022) Nationwide, more than 6,500 households were affected, and insured damages exceeded €145 million (ACA, 2021).

**4.3 Forecast Indicators and Access**

Multiple forecast products were available to national authorities in the lead-up to the July 2021 flood. Forecast outputs signalled a strong likelihood of a high-impact rainfall event several days before the onset of flooding.~~, with signals for a high impact rainfall event emerging several days before onset..~~ From 8 July, ECMWF ensemble precipitation forecasts showed increasing spread and by 12 July, the ensemble mean exceeded the 99th percentile (Magnusson et al., 2021). The Extreme Forecast Index (EFI) for Luxembourg surpassed 0.5 by 9 July and reached 0.8 by 11 July, indicating a very strong signal for extreme rainfall relative to model climatology. This signal remained consistent across successive model cycles. Building on Mohr et al. (2023), who calculated EFI for a larger region mostly covering Germany, we produced values for a $1° \times 1°$ grid box centred on Luxembourg, supporting their findings and adding new insight into Luxembourg-specific EFI evolution. EFI values were derived from ECMWF ensemble forecasts archived in the Severe Event Catalogue (Magnusson, 2019) using ECMWF's operational method, which compares the forecast ensemble distribution to a reforecast-based climatology. Figure 5 shows the daily progression of EFI values, with a steady increase in signal strength over the preceding week.

Deterministic forecasts from ECMWF and MeteoLux did not exceed Luxembourg's national red alert level precipitation thresholds (MeteoLux, 2021). Forecast totals for the Findel reference station remained within the orange level alert range

(*Table 3*). National alert protocols at the time were based on procedural deterministic forecast thresholds and did not include
public facing ensemble-derived indicators such as EFI (Busker et al., 2025).
Forecast access and operational capacity during July 2021 are documented in national user reports and institutional guidance.
MeteoLux and AGE had operational access to ECMWF's IFS/ENS, ICON-D2, ICON-EU, Météo-France ARÔME and
ARPÈGE, and radar composites including RADOLAN (AGE, 2021c; Kobs, 2018). Figure 6 summarises these products,
grouped by type and indicative lead time in 2021. ~~AGE~~ AGE  also operated the Large Area Runoff Simulation Model
(*Landesweiter Flächenhaushalts-Simulationsmodell*, LARSIM), which ingested ensemble and radar-based inputs. Forecasts
were updated every three hours under routine operation and hourly during heightened alert phases. AGE is Luxembourg's
EFAS (European Flood Awareness System) contact point and had access to EFAS outputs during the flood period
(Dieschbourg and Bofferding, 2021; Grimaldi et al., 2023).  No formal EFAS alert was issued, an informal notification for
the Sauer basin was issued at 11:31 on 14 July, less than six hours before peak impacts (Grimaldi et al., 2023; Luxembourg
Government, 2021b). EFAS had issued alerts for the Rhine, Ourthe, Rur, and Moselle from 10 July, but not for Luxembourg
due to dissemination criteria requiring ≥2 000 km² upstream area and persistence across ensemble runs. The internal report
on the flood event stated: *« il reste à préciser que les notifications de l'EFAS sont limitées aux grands fleuves (Moselle, Sûre*
*et Alzette). En aucun cas, les notifications de l'EFAS ne renseignent sur un danger potentiel »* ("it should be noted that
EFAS notifications are limited to the major rivers Moselle, Sûre, and Alzette. In no case do EFAS notifications provide
information on a potential danger") (Luxembourg Government, 2021b).

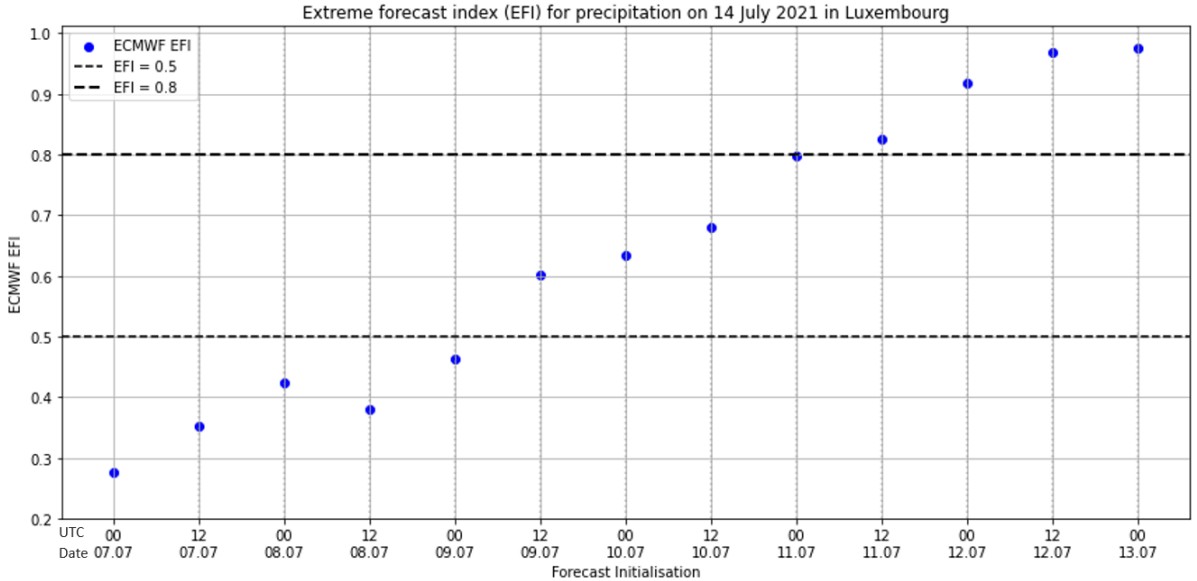


**Figure 5 Progression of ECMWF Extreme Forecast Index (EFI) for 14 July 2021.** Each blue dot shows the EFI value from a different
forecast initialisation between 7 and 13 July. The horizontal dashed lines indicate thresholds of 0.5 (moderate signal) and 0.8 (very strong
signal). EFI values steadily increased over time. indicating high confidence in an extreme rainfall event.

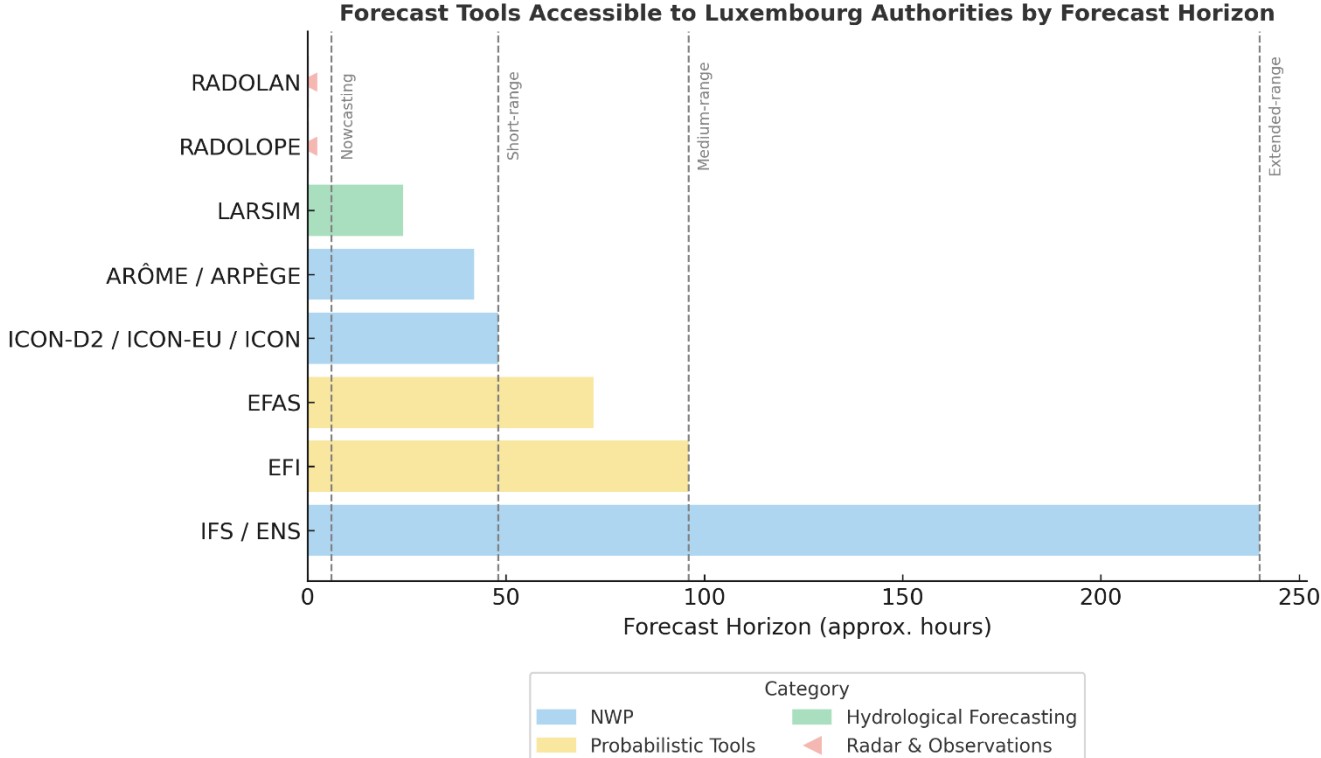


Figure 6 Forecasting products and data sources available to Luxembourg's national meteorological and hydrological authorities (MeteoLux and AGE) during the July 2021 flood event. ~~The table~~This presentation distinguishes between weather and flood-related operational use, grouped by function. Forecast horizons are indicative of standard availability during 2021. This table was compiled from institutional documentation and peer-reviewed literature (AGE, 2021c; Busker et al., 2025; CEMS, 2022; Kobs, 2018; Mohr et al., 2023; Schanze, 2009)

## 4.4 Warning Dissemination Timeline

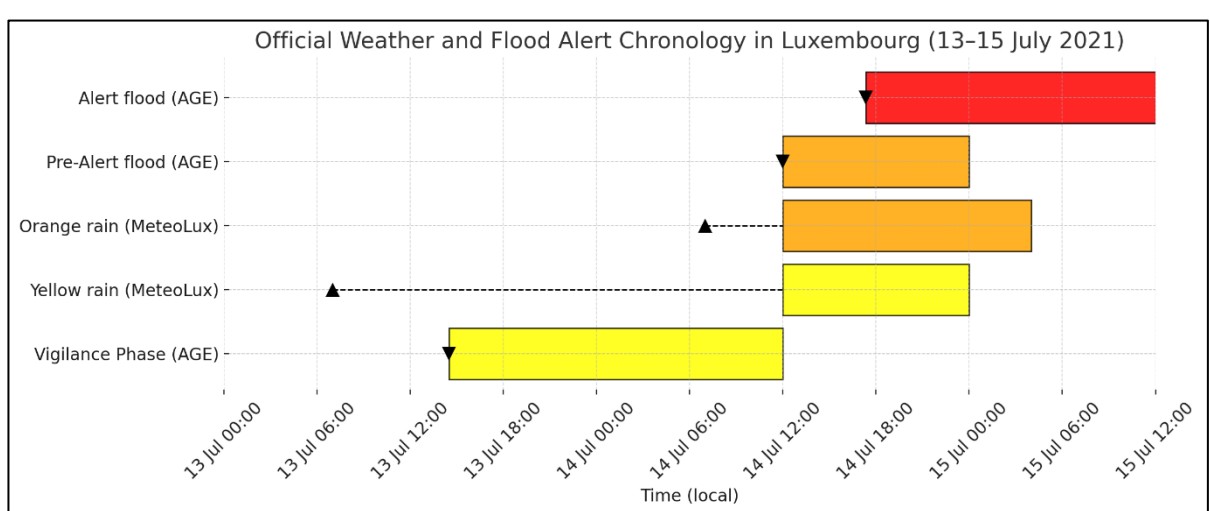

**Figure 7: Official weather and flood alert chronology for Luxembourg, 13–15 July 2021.**
This figure presents the empirical timeline of available forecasts and officially issued alerts during the event Alerts are shown for MeteoLux (weather) and AGE (flood) with triangle markers indicating forecast issuance time and coloured bars representing alert validity periods. Observed impacts and response actions are not represented in this figure. This ~~figure~~chronology is based on official bulletins and institutional records (AGE, 2021a; MeteoLux, 2021a; Gouvernement du Grand-Duché de Luxembourg, 2023).

351

The warning timeline during the July 2021 flood is based primarily on the Luxembourg Government's internal post-event review (Luxembourg Government, 2021b), supplemented by official bulletins from MeteoLux, AGE, and CGDIS, as well as recorded communications and selected media reports. The official warning sequence began on 13 July. At 07:00, MeteoLux issued a yellow alert level rainfall warning, valid from 14 July at 11:00 to 24:00. An orange alert followed at 07:00 on 14 July, valid from 12:00 to 04:00 on 15 July (Luxembourg Government, 2021b). Dissemination occurred via meteolux.lu, inondations.lu email subscriptions and media platforms such as national television broadcaster RTL (www.rtl.lu).

At 14:30 on 13 July, AGE initiated yellow alert level hydrological monitoring for the Sûre, Alzette, Chiers, and Syre basins. On 14 July at 12:00, an orange level flood alert was issued for the southern region, followed by a red alert at 17:20, applicable nationally and valid until 12:00 on 15 July (AGE, 2021a; Luxembourg Government, 2021b; MeteoLux, 2021).

At 14:23 on 14 July, CGDIS sent an informal text message (SMS) to municipal decision-makers, warning of threshold exceedances and encouraging preparatory measures during the orange flood alert. No follow-up text message was issued when the red level flood alert was activated later that day. Behavioural advice was also published on the CGDIS Twitter and Facebook accounts the same afternoon (Biancalana, 2021; CHD, 2021a; Luxembourg Government, 2021b).

Real-time river level updates and flood bulletins were maintained via the website www.inondations.lu. An informal EFAS notification for the Sauer sub-basin was received at 11:31 on 14 July. No formal EFAS alert followed, as ensemble thresholds for basin area and persistence were not met (Dieschbourg and Bofferding, 2021; Grimaldi et al., 2023). No mass notification was issued through the GouvAlert platform.

A national press briefing was held on the afternoon of 15 July and livestreamed through the government portal (Luxembourg Government, 2021a). The Crisis Unit was activated at midnight on 15 July under the Severe Weather Emergency Intervention Plan. According to the government's internal post-event review, this was in accordance with a clause in the Flood Emergency Intervention Plan that assigns flash-flood–type events to the Severe Weather Emergency Intervention Plan. As a result, the activation occurred despite the severe weather alert level remaining at orange, while the flood alert had already reached red earlier that evening. Coordination meetings continued through the night. When the Crisis Unit convened, field-level interventions were already underway. Between 14 and 16 July, CGDIS registered over 8,000 emergency calls to 112 and conducted at least 1,385 recorded interventions. More than 1,600 firefighters, 270 soldiers, and 230 police officers were deployed nationally. The CGDIS coordinated field operations through local fire and rescue stations (*centres d'incendie et de secours,* CIS), focusing on evacuation, public safety, and critical infrastructure protection (CGDIS, 2022; Luxembourg Government, 2021a).

**4.5 Institutional Coordination and Crisis Response**

Coordination at the national level followed the procedures defined in Luxembourg's national emergency intervention framework. The Crisis Unit may be convened following the issuance of a red alert, if conditions meet predefined thresholds concerning urgency, cross-agency coordination and anticipated impact (AGE, 2021a; Luxembourg Government, 2021b). In

accordance with this framework, the Cerisis Uunit was activated by the Prime Minister on the night of 14 July and its first
formal meeting was held at midnight on 15 July, more than six hours after AGE issued a red flood alert at 17:20 (AGE,
2021a; Benoy, 2021; Luxembourg Government, 2021b).
Once active, the Crisis Unit included representatives from MeteoLux, AGE, CGDIS, the Army, the HCPN, the police, and
the Ministry of Home Affairs. Coordination focused on public safety, logistical resourcing, and continuity of operations.
CGDIS and local municipal actors continued to lead evacuation and field logistics. Emergency shelter was provided in
multiple municipalities, and over 560 people were relocated by joint civil-military teams (CGDIS, 2022). Communication
during the peak impact period included updates from multiple agencies via social media, national press and municipal
platforms. A consolidated national bulletin was issued following the activation of the Crisis Unit (Benoy, 2021; CGDIS,
2022; Luxembourg Government, 2021a).

## 396 5. Evaluating Forecast and ~~Warning System~~Early Warning System Performance

**5.1 Comparative Post-Event Evaluation Processes**
Following the July 2021 floods, several European countries conducted formal reviews to assess the performance of forecast
and ~~warning system~~Early Warning Systems. These evaluations varied in scope and method, but shared an emphasis on
institutional transparency and learning. Table 5 summarises the type of reviews conducted, levels of institutional access and
key outputs across five countries. In Germany, technical audits were complemented by parliamentary inquiries in North
Rhine-Westphalia and Rhineland-Palatinate. These revealed major deficiencies in the warning chain, with more than one-
third of surveyed residents reporting that they had not received an alert (BMI and BMF, 2022; Mohr et al., 2023; Thieken et
al., 2023). Cross-country references are included to document procedural arrangements and post-event review mechanisms,
not overall warning system performance.
Belgium's Walloon region initiated an expert-led governance review, resulting in a 146-page report published in
collaboration with the United Nations University Institute on Comparative Regional Integration Studies (UNU-CRIS). A
parliamentary inquiry was proposed but not adopted by the regional government (Lietaer et al., 2024).
In the Netherlands, the Dutch Court of Audit conducted a national review, concluding that warning and evacuation systems
functioned effectively but highlighting the need for improved preparedness and inter-agency coordination. A separate
technical audit by Deltares confirmed the efficacy of warnings in supporting evacuations and recommended more robust
stress testing. Both reviews were complemented by peer-reviewed research outputs (Deltares, 2023; Endendijk et al., 2023;
Netherlands Court of Audit, 2024; Pot et al., 2024).
In France, legally mandated post-event reviews (*retours d'expérience*) on the July 2021 floods were conducted at national
and local levels by the French government. These multi-agency reviews assessed domestic impacts and included analysis of
effects in Belgium, the Netherlands, and Germany. They examined crisis governance, operational coordination, forecasting

and warning, and cross-border cooperation, with findings shared through national channels and via European platforms such as the EU Civil Protection Mechanism (Diederichs et al., 2023).

Unlike neighbouring countries, Luxembourg did not commission an independent or external review of the July 2021 floods. An internal government-led assessment was carried out, but it was not part of any comparative or regional evaluation process. The French government's post-event review notes that requests for information from Luxembourg were either declined or left unanswered (Diederichs et al., 2023; Lietaer et al., 2024). No contributions were made to EU platforms or scientific networks, creating a gap in regional learning.

**Table 3 Comparative post-event evaluation processes following the July 2021 floods.** Review types, parliamentary inquiries, institutional access (as reported in the French government's post-event review unless otherwise noted), key documented outcomes, and publication platforms across five countries (BMI and BMF, 2022; Deltares, 2023; Diederichs et al., 2023; Endendijk et al., 2023; Lietaer et al., 2024; Luxembourg Government, 2021b; Pot et al., 2024)

| Country | Independent Review | Parliamentary Inquiry | Institutional Access to cross-border analysis[3] | Key Review Outcome | Publication Platform(s) |
|---|---|---|---|---|---|
| **Belgium** | Yes Wallonia expert panel | No Inquiry proposed, not adopted | Access granted | 146-page stakeholder-led review; formal inquiry blocked by regional executive | UNU-CRIS (open-access); Regional government portal |
| **Germany** | Yes Technical + stakeholder reviews | Yes NRW and RP state inquiries | Access and cooperation | Surveys: >30% lacked alerts; ~€7 bn in insured losses; two inquiries convened at state level | NHESS journal; State parliament archives; ISF publication (BIH and BF, 2022) |
| **Netherlands** | Yes Deltares technical audit | No | Access granted | Audit confirmed warning efficacy; €455 m in damages; stress testing proposed | Deltares.nl; TU Delft study; PreventionWeb |
| **France** | Yes , post-event review | No | Access granted | Multi-agency learning; findings contributed to EU DRR knowledge-sharing | Ministère de l'Économie portal; EU Civil Protection Forum |
| **Luxembourg** | No, internal review only | No | Access declined, no response | No independent or parliamentary review commissioned | None (no formal publication or participation) |

A standing review mechanism could help address this gap. Such a process could be hosted under the Ministry of Home Affairs and include representatives from MeteoLux, AGE, CGDIS, ASTA, and independent experts. Reviews should be initiated automatically when threshold-impact events occur and examine timelines, institutional coordination, and communication processes. Without a formal structure for review, lessons remain anecdotal and preparedness does not evolve.

---

[3] Refers to the degree of cooperation and information-sharing with the French government's legally mandated post-event review (*retour d'expérience*), which included cross-border analysis of the July 2021 floods in Belgium, the Netherlands, Germany, and Luxembourg.

**5.2 Why Forecasts Did Not Lead to Action**

Forecast guidance in the days leading up to the July 2021 flood presented clear signals of extreme rainfall and pointed to a statistically rare and potentially high-impact rainfall event (Mohr et al., 2023; Thompson et al., 2025). However, Luxembourg's national ~~warning~~ alert level did not move beyond yellow until the morning of 14 July. In the days immediately preceding the flooding, institutional interpretation was based primarily on deterministic rainfall totals at the Luxembourg-Findel reference station, where forecast and observed precipitation remained below the national red alert level procedural threshold (MeteoLux, 2021; Ministry of State et al., 2015). Observations from other stations, in central and northern Luxembourg exceeded these red alert-level criteria, but these sites were not included in the formal decision-making protocol (AGE, 2021c; HCPN, 2019; Szönyi et al., 2022). Ensemble indicators, while reviewed internally, had no procedural role in alert level decisions (Busker et al., 2025).

Forecast skill was not the limiting factor. Forecast products from ECMWF, ICON-EU, and Météo-France consistently showed elevated rainfall potential across the wider region (Mohr et al., 2023; Thompson et al., 2025). Several ensemble members projected accumulations well above the return periods typically used in warning calibration. At the time, however, there was no mechanism in national procedures to translate these probabilistic signals into operational triggers for alert escalation or plan activation. The protocol relied on thresholds applied to a single reference station, with no formal post-processing of ensemble outputs.

Hydrological forecasts showed a similar pattern (Busker et al., 2025; Montanari et al., 2024). Although AGE used ensemble and radar-based inputs within the LARSIM model, public bulletins were deterministic, and probabilistic information was not formally linked to warning alert level changes (Busker et al., 2025).

Public communication during this period reflected the same deterministic framing (Zander et al., 2023). On the evening of 13 July, RTL's national news broadcast quoted MeteoLux:

 "*From Wednesday morning until Thursday, larger amounts of rainfall could reach us, so we need to be a bit cautious.*"

 The presenter added:

 "*Foreign weather services are talking about 100 litres per square metre, but for Luxembourg, the warning levels are still only at yellow.*" (RTL, 2021a)

This comparison emphasised that while neighbouring services, including in directly connected catchments, were warning of extreme totals across the border, Luxembourg's own alerts remained in the yellow range (below 31 mm in six hours or 51 mm in 24 hours). No reference was made to EFI values or to the consistent ensemble signals emerging across multiple models. The first orange level rainfall warning was issued on the morning of 14 July and took effect at 12:00, after heavy rain had already begun in parts of the country (AGE, 2021a)

The Prime Minister's public statement after the event reinforced the framing of the flood as unexpected.

 *"No one could have predicted the extent of the flooding as it unfolded in mid-July, and it was nothing short of a miracle that no one had been seriously harmed by the catastrophe."* (RTL, 2021b)

While precise local impacts could not have been forecast with certainty, the broader signal of an extreme rainfall event had
been evident in ensemble guidance for several days. The challenge was the absence of institutional mechanisms to interpret
and act on probabilistic signals under uncertainty.
One recommendation would be to formally integrate probabilistic forecast tools such as the Extreme Forecast Index (EFI)
within national warning protocols when converging probabilistic signals indicate the potential for severe impacts (Busker et
al., 2025; Cloke and Pappenberger, 2009; Mohr et al., 2023). Ensemble outputs should be post-processed into operational
scenarios and supported by targeted training. Observational data from ASTA and municipal networks should also be
integrated when they exceed warning criteria (Lanfranconi et al., 2024; Szönyi et al., 2022). These measures would support
earlier action when risk is emerging, rather than only after it is confirmed by deterministic indicators.

**5.3 How Thresholds Delayed the Response**

Luxembourg's warning protocols were structured around fixed procedural rainfall thresholds measured at a single reference
station. Under the Severe Weather Emergency Intervention Plan, a red level weather warning may be issued if rainfall
exceeds 80 mm in 24 hours or 45 mm in 6 hours at the Luxembourg-Findel station (HCPN Law, 2016). On 14 July, Findel
recorded 74.2 mm over 12 hours, breaking its all-time daily record for any month since observations began in 1947, yet no
red alert level warning ~~level warning~~ was issued (MeteoLux, 2021)
Other stations from the ASTA network also recorded totals above red-~~level~~colour-coded alert level criteria on 14-15 July
(AGE, 2021a). These observations were not included in the formal warning framework and therefore played no role in real-
time decision-making (HCPN, 2019). Excluding a large share of the available observational network from official warning
protocols is not unique to Luxembourg and has been identified in other regions that rely on narrowly defined deterministic
systems (Cosson et al., 2024; Trošelj et al., 2023).
This arrangement created a structural limitation. The agrometeorological network operated by ASTA includes over 35
weather stations across the country. However, institutions did not recognise their data within the official warning framework
(HCPN Law, 2016). Consequently, a significant share of Luxembourg's observational infrastructure was excluded from the
official process of warning generation.
Hydrological forecasting faced similar structural constraints. The use of probabilistic inputs was limited to internal
processing and no mechanisms were in place for using this information to support escalation to higher colour-coded alert
levels. ~~in operational warning escalation~~ (Busker et al., 2025; Haag et al., 2022).
~~Threshold~~Procedural thresholds defined when warnings could be issued~~, but also~~and the basis on which decisions were
deemed valid. In theory, the presence of a single institutional threshold at Findel was meant to simplify decisions. In
practice, it constrained them. Even when that station recorded historically extreme rainfall, no warning level change
followed. In neighbouring countries, Early Warning Systems operated under different procedural criteria, allowing alert
decisions to draw on exceedance across regional observation networks and convergence within ensemble forecast products,
leading to earlier issuance of red-level alerts on 13 July.~~Neighbouring countries responded differently. Germany and~~
~~Belgium issued red alerts on 13 July, one day earlier, based on consistent observational exceedance across regional networks~~
~~and convergence within ensemble forecast products.~~ Their approaches allowed for distributed decision-making using broader
spatial criteria, rather than relying on one location to validate action (Lietaer et al., 2024; Mohr et al., 2023).
A key recommendation is to formally integrate Luxembourg's existing observational infrastructure such as ASTA stations
into the operational ~~warning system~~Early Warning System, allowing wider spatial validation of hazard signals. A
parliamentary question in July 2024 proposed merging Luxembourg's two public meteorological services to improve
efficiency and integration. The government confirmed that while discussions had been held since 2018, the proposal was not
adopted. It stated that cooperation between MeteoLux and ASTA had been sufficient and that the implementation of LU-
Alert provided a direct channel for transmitting official warnings to the public. On this basis, it argued that a merger was
unnecessary and confirmed that no such measure was foreseen in the 2023–2028 government programme (CHD, 2024).
However, no evidence was presented on how this arrangement addresses the structural limitations identified in the July 2021
event. In parallel, AGE should implement probabilistic flood forecasting workflows that carry procedural weight. These
steps would increase situational awareness and reduce dependence on a single reference station (Ebert et al., 2023; Golding,
2022; WMO, 2024b).

**5.4 When Warnings Did Not Reach the Public**
During the July 2021 flood, Luxembourg's public alerting systems were not used in a way that enabled timely ~~early~~
protective action. The GouvAlert mobile application, designed to send real-time emergency notifications, did not transmit
any message on 14 July. A scheduled alert was not delivered due to an expired Secure Sockets Layer (SSL) certificate, and
no warning reached users during the hours when rainfall intensified and river levels began to rise (Tobias, 2021).
Institutional communication remained limited. At 14:23 on 14 July, CGDIS issued an SMS to local authorities referencing
orange-level conditions. The message did not contain the word "alert" and was not accompanied by a wider public advisory
(CHD, 2021b; Luxembourg Government, 2021b). No coordinated national message was issued through press channels or
social media before flood impacts were widely reported. Infocrise.lu, which serves as the government's official crisis
information portal, is not designed to function as a real-time alerting tool and was not used for that purpose during the
warning phase (HCPN Law, 2016). The communication environment during the flood evolved across multiple platforms,
with limited coordination prior to impact.
Multilingual accessibility may also have limited the reach of warning messages. Luxembourg's official languages are
Luxembourgish, French, and German, but alerts are often issued in one or two languages only. (STATEC, 2022) estimates
that only around 60 percent of the population speaks Luxembourgish fluently. Many residents rely on French or German for
official communication, and a significant proportion of the workforce consists of daily cross-border commuters. In this
context, the absence of standardised multilingual communication protocols can reduce the effectiveness of public alerts,
particularly in linguistically diverse populations (Hannes et al., 2024; IFRC, 2020; Kalogiannidis et al., 2025; UNDRR,
534 2022)

While these issues were not the primary cause of limited operational response during the flood, they revealed how dependent
the system had become on a small number of delivery channels. This became evident on 16 July, when the MeteoLux
website went offline due to a server failure and remained inaccessible until 19 July. During this period, CGDIS continued
referring the public to the offline site (Tobias, 2021), highlighting a lack of contingency planning for communication
continuity (Reichstein et al., 2025).
Following the flood, Luxembourg introduced LU-Alert, a multilingual cell broadcast system designed to deliver real-time
notifications to all mobile phones in a given area. While this improves technical capacity, it does not resolve the procedural
barriers that limited alert use in July 2021. Without clearly defined protocols for who authorises and triggers alerts, when,
and through which channels, even advanced systems may fail to support timely action (Oliver-Smith, 2018; WMO, 2022).
The 2024 DANA floods in Valencia illustrate how procedural communication choices, including alert timing and message
content, can limit the protective value of public warnings. The 2024 DANA floods in Valencia illustrate this challenge.
Spain's ES-Alert system functioned technically, but alerts were issued at a stage in the event when opportunities to influence
public decision-making were already reduced. Post-event reviews linked this to weak integration between forecast
interpretation and operational decision-making (Aznar-Crespo et al., 2024; Galvez-Hernandez et al., 2025; Martin-Moreno
and Garcia-Lopez, 2025). Luxembourg faces similar risks if alert systems remain detached from institutional procedures.
Effective public communication requires more than new infrastructure. A central protocol should define when alerts are
triggered, which institutions are responsible, how content is translated across platforms and languages, and how redundancy
is ensured. Without these structural measures, warnings may not reach the public in time to support protective action.
In the national system, warnings are intended to reach residents through official dissemination channels, including public
alerting systems, press communication, and institutional information platforms. This analysis focuses on the institutional
conditions that shape whether public warnings can be authorised, issued, and disseminated, rather than on how residents
interpret or respond to those warnings.

## 5.54 Coordination Only Began After Impact

Luxembourg's emergency coordination during the July 2021 flood was constrained by a procedural sequence that delayed
strategic activation. Although flood forecasts and operational responses were already active on 14 July, national-level
coordination through the Crisis Unit was only initiated at midnight, several hours after widespread flooding had begun. This
delay stemmed from a rigid stepwise process, a red alert had to be issued, followed by a ministerial evaluation, before cross-
agency coordination could be formally launched (CHD, 2021a; Luxembourg Government, 2021b).
Operational agencies, including CGDIS, MeteoLux, and AGE, responded to early signals. CGDIS alone handled over 1,200
calls and deployed more than 100 units throughout the day (CGDIS, 2022). However, without formal activation of the Crisis
Unit, no unified public messaging or strategic coordination was possible. Communication remained decentralised and limited
to agency-specific channels.
This misalignment occurred despite the existence of both capacity and legal authority. It reflected procedural inflexibility
that prevented early convergence of information and action. As highlighted by (Hegger et al., 2016), effective flood risk
governance requires both anticipatory mechanisms and coordination structures that can adapt in real time. In fast-onset
crises, ~~formal threshold~~procedural thresholds may delay the shift from proactive intervention to reactive response (Lietaer et
al., 2024).
To improve future alignment, Luxembourg could revise procedural thresholds to enable early coordination based on
consistent forecast indicators, such as rising hydrometric levels and multi-agency consensus. A shared operational platform
involving AGE, MeteoLux, CGDIS, and crisis managers could allow joint interpretation of dynamic risks, enabling earlier
activation even before red alert level thresholds are formally crossed (Amarnath et al., 2023; Dasgupta et al., 2025; Šakić
Trogrlić and Van Den Homberg, 2022). This would help ensure that national-level coordination begins in response to
emerging risk, rather than observed impacts.

**5.6~~5~~ Reading Forecasts as Policy Signals**
Forecasts ahead of the July 2021 flood contained multiple early indicators of an emerging regional hazard. EFI values
exceeded 0.8 by 11 July, and EFAS issued alerts for nearby river basins from 10 July onward. These signals, documented in
~~widely recognised in~~ post-event evaluations in Germany and Belgium, were also available in Luxembourg, but they did not
inform operational decision-making (Lietaer et al., 2024; Mohr et al., 2023).
Although EFAS and EFI were monitored internally by AGE and MeteoLux, no procedural framework existed in
Luxembourg for ~~acting on~~ using these products to inform warning level decisions and public communication. In Germany
and Belgium, post-event analyses describe procedural arrangements that allowed ensemble-based and regional information
to be considered within warning processes, without implying more effective outcomes. In Luxembourg, the absence of an
equivalent framework meant that these forecasts remained outside formal decision pathways and no institutional review has
clarified how such inputs could be interpreted or integrated. ~~Unlike Germany and Belgium, Luxembourg did not use these~~
~~forecasts to justify public warnings, and no institutional review has clarified how such inputs should be interpreted or~~
~~integrated.~~ EFAS alerts, while designed for larger river systems, still provide contextually valuable information, especially
when interpreted alongside local data. Treating them as irrelevant, rather than evaluating their limitations constructively,
limits the system's ability to recognise transboundary risk (Busker et al., 2025; Mohr et al., 2023).
The problem is not the forecasts, but the absence of structures to interpret and act on them collectively. Luxembourg's
warning framework remains tied to deterministic ~~threshold~~procedural thresholds without a mechanism for incorporating
probabilistic guidance. EFI and EFAS are treated as reference data rather than operational tools and their signals hold no
procedural weight.
It is recommended that Luxembourg establish a formal joint interpretation mechanism involving MeteoLux, AGE, CGDIS,
and other relevant actors, to review ensemble guidance and translate it into operational scenarios. This process would allow
for expert judgement to be exercised under uncertainty and would increase the policy relevance of probabilistic signals
(Hoffmann et al., 2023; WMO, 2024b). Forecasts can support anticipatory action, but only if the system is configured to read
them as policy-relevant signals, not technical background.

# 6 Risk Interpretation and System Structure

~~Early warning systems~~Early Warning Systems are widely recognised as central to disaster risk reduction (Kelman and
Glantz, 2014; Šakić Trogrlić and Van Den Homberg, 2022; UNDRR, 2015; WMO, 2024b). They are typically embedded in
frameworks that conceptualise disasters into sequential phases of preparedness, response, recovery, and ~~mitigation~~risk
reduction. These phases are often assumed to unfold in a linear progression, with decisions and responsibilities evolving
predictably over time (Berke et al., 1993; McEntire, 2021). However, critical perspectives challenge this view, emphasising
that disasters emerge within complex, uncertain, and structurally constrained systems (McDermott et al., 2022; Wilkinson,
612 2012).
The analysis builds on those insights by examining how institutional structures shape the interpretation of risk. It introduces
the Waterdrop Model and applies it to the July 2021 floods in Luxembourg.

**6.1 The Waterdrop Model**

The Waterdrop Model is a structural model for analysing how Early Warning Systems filter risk signals (Figure 8). The
model was developed as a diagnostic extension of the reconstructed value chain and is intended to examine how institutional
design conditions the use of forecast information and is not intended as a prescriptive or deterministic framework. Developed
through reflection on the Luxembourg 2021 flood, the model builds on the value chain approach by clarifying how
institutional configuration not just communication or technical capacity determines whether forecast information can lead to
anticipatory action (Cloke and Pappenberger, 2009; Hermans et al., 2022; Golding, 2022). Rather than assuming that signals
automatically translate into ~~response~~action, the model helps identify how value is conditioned by the system into which
information enters. The Waterdrop Model explains how mandates and responsibilities shape whether forecast information
can lead to action.
Figures 7 and 8 are intended to be read together, with Figure 7 documenting the empirical sequence of forecasts and alerts
during the event and Figure 8 providing a conceptual framework for interpreting how the warning system processed that
information.
At the centre of the model is a triangle representing the architecture of a national ~~warning system~~Early Warning System.
Each corner of the triangle corresponds to ~~one of three gatekeeping elements;~~ authorised data sources, predefined procedural
thresholds and designated institutional mandates that define responsibility and the timing of warning authorisation and
dissemination. Only when a signal passes through all three originating from a recognised source, exceeding a defined
~~threshold~~procedural threshold, and falling within the responsibility of an authorised actor can it initiate protective measures
(Alfieri et al., 2012; Antwi-Agyakwa et al., 2023). These thresholds function as institutional decision rules that intersect with
governance arrangements by defining when responsibility shifts from monitoring to authorisation and action. If any of these
conditions are not met, the signal may circulate informally but cannot trigger official warning. The triangle defines the
system's operational boundaries for action. Within this structure, procedural bottlenecks can delay escalation and
dissemination even when risk information is available and technically credible.
Surrounding this core are institutional actors, forecast and data products, observational networks that may hold operational
relevance but lack formal standing within the warning protocol. These include probabilistic forecast products, transboundary
alerts, local data sources, and expert assessments from actors without decision authority. The model distinguishes between
signals that are visible and those that are usable within institutional procedure (De Coning et al., 2015; Jaime et al., 2022).
Information may be available, but it only becomes actionable when it meets the system's internally defined criteria.

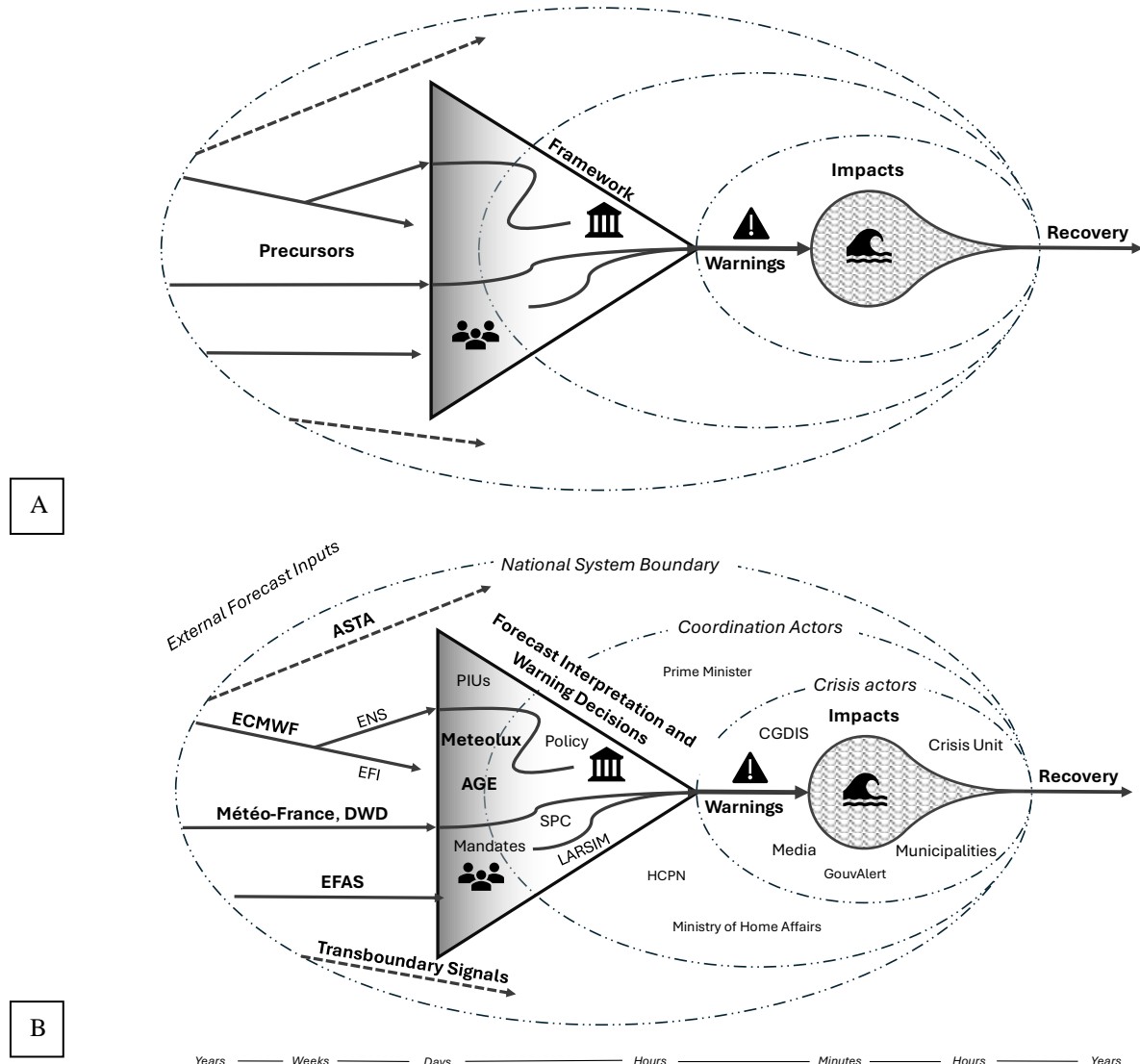

**Figure 8 The Waterdrop Model:** How Structural Design Filters Risk Information in Early Warning Systems. This figure provides a conceptual representation of how warning systems process and filter forecast information.

***Panel A*** presents the conceptual model. Forecast signals enter from the left and are filtered through a triangular warning core defined by three structural components: authorised data sources (left corner), procedural thresholds and policy rules (top), and institutional mandates (bottom). Only signals meeting all three criteria progress to warnings and response. Dashed arrows represent excluded signals. Concentric ellipses represent the narrowing opportunity for anticipatory action, aligned with the timeline at the base.

***Panel B*** applies the model to Luxembourg's 2021 flood. Forecast inputs from ECMWF, EFI, EFAS, ASTA, and cross-border sources were available but remained outside national procedures. Only deterministic inputs from authorised actors (MeteoLux, AGE) passed through the system's triangle via PIUs and LARSIM. Signals lacking procedural status were filtered out. On the right, warnings connect to coordination actors (CGDIS, municipalities, Crisis Unit), with post-warning actions and impacts shown. The national system boundary illustrates how institutional design limited the use of probabilistic and transboundary signals.

A timeline at the base of the model represents the narrowing window for anticipatory action as a ~~hazard~~hazard event evolves.
As time passes and certainty increases, more signals may enter the triangle, but the decision space for anticipatory action
narrows, increasing the risk that warnings are issued once impacts are already unfolding. ~~As time passes and certainty~~
~~increases, more signals may enter the triangle, but the opportunity for mitigation diminishes.~~ The model is intentionally
diagnostic. It does not propose an ideal structure, but instead clarifies how institutional design choices govern the use of
information. It supports critical analysis of how systems configured around deterministic certainty and linear authority may
fail to act on probabilistic or emerging risk, even when warnings are technically available (Arnal et al., 2020; Bouttier and
Marchal, 2024). In Luxembourg, this structural filtering was reinforced by reliance on a single reference station at Findel,
which concentrated procedural authority in one location and increased vulnerability to delayed threshold exceedance

**6.2 Application to the Luxembourg 2021 Flood Disaster**
The Waterdrop Model helps explain why Luxembourg's national ~~warning system~~Early Warning System did not activate
early action in response to multiple early indicators of flood risk in July 2021. Ensemble forecasts from ECMWF, EFI values
exceeding 0.8, and EFAS alerts for neighbouring basins all pointed to a high-impact rainfall event. No warning level
increase occurred until deterministic thresholds were breached, and national coordination began only after widespread
impacts were already underway (Busker et al., 2025; Haag et al., 2022). This outcome was not due to a lack of forecast
capacity, but to the system's structural configuration.
Under Luxembourg's operational rules, meteorological warnings could only be issued by MeteoLux on the basis of
deterministic forecasts from the Findel reference station, while hydrological alerts from AGE depended on observed
exceedance at designated gauging stations. Forecasts from ensemble systems, Extreme Forecast Index values, EFAS alerts,
and observations from other networks such as ASTA's agrometeorological stations were available but held no formal status
within the Weather and Flood Emergency Plans (Section 3). These products could inform internal situational awareness, but
they were not recognised as valid inputs for official activation or public warning.
Although AGE had access to probabilistic flood forecasts and ensemble precipitation inputs through models such as
LARSIM, these were not operationalised in the alerting process. As noted in (Busker et al., 2025), probabilistic outputs are
used internally but have no procedural consequence. The national ~~warning system~~Early Warning System was designed to act
on deterministic exceedance at specified locations, not on converging probabilistic evidence. Even when credible signals
were identified, there was no mechanism to translate those signals into formal decisions unless they matched the authorised
criteria embedded in national protocol (Jaime et al., 2022).
This design filtered out signals that were visible but procedurally unusable. Despite record precipitation at Findel and
extreme rainfall recorded at other stations, no procedural mechanism existed to escalate warnings based on broader
observational or probabilistic evidence. Godbrange recorded over 100 mm of rainfall in 24 hours, well above the red alert
level threshold but this observation played no role in national activation because it came from a station not designated in the
Emergency Plan. EFAS alerts issued for upstream river basins in Germany and Belgium were not extended to Luxembourg
due to dissemination criteria that required a minimum upstream catchment area of 2000 km² and persistence across multiple
ensemble cycles. An informal notification for the Sauer was received shortly before peak impacts but held no formal status.
Forecast interpretation remained tied to deterministic exceedance from nationally authorised sources.
Coordination followed the same logic. The emergency protocols allow for the convening of an inter-institutional Evaluation
Cell during orange or red alert phases. This unit, chaired by the responsible technical authority (Meteolux or AGE), assesses
conditions and advises the HCPN on whether national coordination is required. However, activation of the Crisis Unit
remains a political decision and must be authorised by the Prime Minister. In July 2021, this process delayed formal cross-
agency coordination until midnight on 15 July, by which time widespread impacts were already unfolding (Hagenlocher et
al., 2023). No procedural mechanism existed to initiate anticipatory coordination based on converging probabilistic signals.
The system remained in observation mode until deterministic ~~threshold~~procedural thresholds were exceeded.
The Waterdrop Model captures this disconnect. It shows how system structure rather than technical capacity determined
what information could lead to action. In Luxembourg, early signals were present, but action was delayed not necessarily
because they were missed, but because they were procedurally unusable. The model highlights how protocols that prioritise
deterministic certainty and formal authority may struggle to respond under uncertainty, even when forecasts provide advance
warning.

**6.3 Implications for Systemic Risk and Governance**
The 2021 flood disaster illustrates how ~~early warning systems~~Early Warning Systems can be technically capable but
structurally restricted. Convergent and credible risk information was available, but the system design prevented early action
based on early warning. The Waterdrop Model shows that these dynamics emerge not from isolated misjudgements, but
from how institutional arrangements define valid inputs and allocate authority to respond (Kelman and Glantz, 2014; Oliver-
Smith, 2018).
Systems that rely heavily on fixed thresholds, sequential decision-making processes and limited incorporation of
probabilistic signals may systematically exclude useful early indicators. These systems are optimised for certainty, not for
emerging or partial information. As a result, action may only begin once impacts are visible, reducing forecast value and
shortening the response window (Šakić Trogrlić and Van Den Homberg, 2022).
The absence of a formal post-event review in Luxembourg suggest how governance cultures shape system learning. While
several European countries initiated independent evaluations following the 2021 floods, Luxembourg did not. This suggests
a governance context where formal post-event review is not institutionalised as standard practice.
Technical upgrades alone cannot resolve these challenges. The launch of LU-Alert improved message delivery capacity, but
the limitations observed in 2021 were primarily structural.
How institutions handle uncertainty also shapes trust in warning systems. When uncertainty is communicated implicitly
through procedural delay or conservative escalation, it may weaken confidence among both officials and the public.
Repeated exposure to warnings that do not lead to visible action further raises communication and risk-education challenges,
for decision-makers tasked with interpreting evolving signals under uncertainty.

The Waterdrop Model highlights how systemic risk can emerge not only from external hazards, but from internal design features of governance systems. This reflects a broader understanding of systemic risk as emerging from the structure and configuration of ~~warning system~~Early Warning Systems themselves (Bosher et al., 2021; Golding, 2022; Šakić Trogrlić and Van Den Homberg, 2022). These insights align with critical analyses of disaster governance that emphasise how institutional design filters what counts as actionable information (Alcántara-Ayala and Oliver-Smith, 2016; McDermott et al., 2022; Wilkinson, 2012). It highlights how the operational value of information depends on whether systems are configured to use it. While effective early warning depends on whether warnings are understood and acted upon by residents, this analysis focuses on the institutional conditions that determine whether such warnings can be authorised, escalated, and disseminated in the first place. The design, targeting, and evaluation of resident-facing messages are therefore recognised as essential, but lie beyond the empirical scope of this study. A more detailed mapping of domain-specific processes and interpretive practices within institutions would require data beyond those available for this analysis and represents a priority direction for future research.

Early Warning Systems are not only about detecting hazard signals. They are about whether institutional structures enable interpretation and coordinated action in time. Without that capacity, even the most advanced forecast systems may struggle to prevent disaster.

This analysis is based on publicly available records, institutional documentation, and reconstructed timelines, and is therefore limited to formally documented procedures, mandates, and authorised communication channels within the national warning system. Informal decision-making, undocumented interpretations, and internal deliberations are not captured. The analysis further focuses on the warning system up to the point at which alerts are issued to the public. Public interpretation, behavioural response, and message effectiveness are not examined, as these dimensions require different data and methods. These limitations should be considered when interpreting the findings. They also highlight an important direction for future research on people-centred early warning, in which institutional analysis is complemented by studies of public understanding and response.

## 7. Conclusion

Early Warning Systems are widely recognized as essential tools for disaster risk reduction. As ~~we~~ demonstrated by severe floods of July 2021 in Luxembourg, having forecast information available does not guarantee that early action will follow. While forecast signals were available several days in advance, procedural systems prioritised action based on confirmation rather than forecast-based uncertainty. Using a value chain approach, we traced how forecast information moved through Luxembourg's ~~warning system~~Early Warning System and identified points where timing, procedural thresholds, and divided responsibilities limited anticipatory action. These constraints were not caused by inaccurate forecasts but by how risk information was understood, prioritised, and ~~acted upon~~translated into action within existing structures.

To support this analysis, the Waterdrop Model was introduced to show how forecast signals interact with institutional rules
and operational timelines. It clarifies why credible early indicators may not lead to timely decisions when systems depend on
predefined criteria or rigid procedural steps. The model also highlights how time pressure and fragmented responsibilities
can hinder collective interpretation, especially when institutions lack not only authority but also the resources and structures
needed to act on probabilistic guidance.
Luxembourg's experience reflects a broader challenge. An effective ~~warning system~~Early Warning System derives its value
from the capacity of institutions to interpret forecasts as actionable signals and to mobilise timely, coordinated responses
under uncertainty. The analysis returns to the central question of how forecast signals were translated into anticipatory action
during the July 2021 floods in Luxembourg. The findings show that institutional design largely determined whether early
information could be authorised, interpreted, and acted upon in time. The Value Chain approach and the Waterdrop Model
show how governance structures shape the operational value of forecasts in Early Warning Systems across different
institutional settings.
**Author Contributions**
JDC led the investigation, conducted the analysis, and wrote the manuscript as part of his PhD research. EE and DH
contributed to the development of the value chain framework and the design of the database questionnaire. HLC and JN
supervised the PhD project and provided conceptual guidance and feedback on the manuscript. All authors contributed to
discussions of the results and approved the final version of the paper.
**Acknowledgements**
This research was conducted as part of JDC's PhD at the University of Reading. The author thanks HLC and JN for
supervision and guidance throughout the project, and EE and DH for their collaboration, input, and support. The author also
acknowledges the foundational work of the WMO WWRP HIWeather Value Chain Project team, and the continuity of that
work supported by the UCL Warning Research Centre, which hosts the value chain questionnaire used in this study.
Additional thanks go to Linus Magnusson (ECMWF) and Thomas Schreiner (ESSL) for providing access to forecast and
severe weather data used in the analysis.

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
