# Peer review of "Signals Without Action: A Value Chain Analysis of 2 Luxembourg's 2021 Flood Disaster"

_EGUsphere, 2025_

## Author Response (AR1)

**POINT-BY-POINT RESPONSE TO THE REVIEWS AND LIST OF RELEVANT CHANGES**

**GENERAL RESPONSE TO ALL REVIEWER AND COMMUNITY COMMENTS**

**1. General, terminology and conceptual consistency**
- "Early Warning System" is now used consistently when referring to institutional systems, while "early warning" refers to the broader process.
- Typos and punctuation has been corrected throughout the manuscript.
- Numbering of headings etc has been corrected and streamlined as well.
- The term "hazard" has been replaced with "hazard event" where an imminent or unfolding event is meant.
- "Mitigation" has been replaced with "risk reduction" in disaster-risk contexts, to avoid confusion with climate-mitigation terminology.
- Alert levels are consistently described as colour-coded, and thresholds are explicitly framed as procedural and institutional.
- Institutional roles are clearly distinguished between issuing forecasts and warnings and activating emergency plans or crisis procedures.

**2. People-centred framing**
The manuscript now clarifies that the analysis focuses on institutional processes, while recognising that these processes shape whether warnings can reach and support resident protective action.

**3. Structural**
- A short bridging paragraph has been added near the start of Section 1 to preview Luxembourg's early warning and emergency governance system.
- The July 2021 floods are now explicitly described as having been formally declared a 'natural' disaster in Luxembourg.
- The regional context has been completed by explicitly noting that France was also affected by the July 2021 floods.

**5. Abstract clarification**
- The abstract now clarifies that some action did occur, but that it was not early or protective enough to prevent major impacts.

**6. Conceptual clarification of the Waterdrop Model**
- The Waterdrop Model is now explicitly described as:
  - Developed for this study
  - A diagnostic extension of the reconstructed value chain,
  - Non-prescriptive and non-deterministic.
- A concise analytical statement has been added to clarify the model's added value in explaining how mandates and responsibilities shape whether forecast information can lead to action.

**7. Figures and tables**
- Figure 4 caption now explicitly explains all symbols and reference lines.
- Figures 7 and 8 are now clearly distinguished:
  - Figure 7 as the empirical forecast and alert timeline,
  - Figure 8 as a conceptual representation of system operation.

**8. Methodological limitations**
- A short methodological limitations paragraph has been added to the emd of the discussion section.

Please note that Line changes may have occurred during formatting etc, they are indicative and only most relevant changes are quotes in the responses that follow. Please see track changes for full changes done to the manuscript. Thank you.

**DETAILED RESPONSE TO REVIEWER AND COMMUNITY COMMENTS INCLUDING LIST OF RELEVANT CHANGES**

**RC1 REVIEWER REPLY INCLUDING LIST OF RELEVANT CHANGES**

Thank you for taking the time to review our manuscript and the really useful feedback provided!

Below is the detailed reply (Author Reply) to each of the points raised in previous RC1 comments (**RC1: bold**):

> **RC1: Generally, this is a very interesting subject well presented and discussed.**
>
> **RC1: The text can be improved by additional consistency of key hazard and risk terms and using them also more consistenty.  I am surprised you don't introduce EWS as an acronym since you mention the long term Early Warning System so often.**

**Author Reply:** Thank you for this observation. The decision not to use the acronym EWS was a choice, as the paper aims to remain accessible to both technical and policy audiences. Writing out Early Warning System in full was a choice intended to support readability across disciplines.

> **RC1: Structurally, a weakness in the manuscript can be overcome by consequently introducing every concept and key aspect first before discussing it. You confuse the reader by already discussing warning levels, thresholds and apps and mechanisms before they are properly introduced. This should be done by adding an earlier section on "Early Warning Systems and its governance in Luxembourg" or similar. I would start with the paragraph line 115 as the motivation of your work, then outline the Luxembourg EWS and emergency governance system as suggested above, then continue with current line 110.  You currently have the challenge that you describe this in more detail in section 3.1 but this comes too late.**

**Author Reply:**  The current sequencing was chosen to introduce the broader conceptual and transboundary context before presenting national governance details. Section 3.1 was placed after the methodological framework to maintain a consistent transition from international to national scales of analysis. We recognise, however, that introducing the structure of Luxembourg's Early Warning and emergency governance system earlier could improve reader orientation. In a future revision, we plan to add a short bridging paragraph near the start of Section 1 to preview this framework while keeping Section 3.1 as the detailed reference.

**List of relevant changes:**

**Lines 67-71: '**In Luxembourg, early warning and emergency management are organised within a centralised national governance system, with no intermediate regional tier between national authorities and municipalities. Forecasting, warning issuance, emergency planning, and crisis coordination are assigned to distinct national institutions operating under formal procedures and predefined thresholds. The following sections introduce the national and transboundary context of the July 2021 floods, while Section 3 provides a detailed description of institutional roles, responsibilities, and activation protocols.**'**

**RC1: Conceptually, I am surprised that in today's focus on "people centered EWS" and the action by the people who need to be safe from the hazard events, neither the established value chain approach nor your work really focuses on that elemental, "first mile" aspect of the people itself but stops with "official decision-makers". Ultimately, the decision-maker is the individual, household head or community that can or cannot keep safe. I would have liked to see this discussed more. See https://www.sciencedirect.com/science/article/pii/S2589004225006145.**

**Author Reply:** Thank you for this comment. We examine how forecasts and warnings were handled within Luxembourg's national warning system to explain why early signals did not lead to better anticipatory action. We completely agree that the "first mile" is where early warning must succeed. While this paper focuses on institutional processes, these determine whether people receive clear and actionable messages. The findings therefore help to explain why the link between official warnings and public action unfolded in the way it did. Direct community responses were beyond the available data. We share the view expressed by Budimir et al. (2025) that people-centred early warning requires clear connections across all layers of the system. In the revised discussion we will note this explicitly and position Luxembourg's structural barriers as one factor limiting that connection.

**List of relevant changes:**

**Lines 728-735:** 'This analysis is based on publicly available records, institutional documentation, and reconstructed timelines, and is therefore limited to formally documented procedures, mandates, and authorised communication channels within the national warning system. Informal decision-making, undocumented interpretations, and internal deliberations are not captured. The analysis further focuses on the warning system up to the point at which alerts are issued to the public. Public interpretation, behavioural response, and message effectiveness are not examined, as these dimensions require different data and methods. These limitations should be considered when interpreting the findings. They also highlight an important direction for future research on people-centred early warning, in which institutional analysis is complemented by studies of public understanding and response.'
* * *
**RC1: With regards to uncertainty, while I agree with your discussion in 5.2, it requires some further thinking - this is not just about integrating uncertainty into the forecast and how it might trigger higher alert levels, but then also how laypeople both in official functions as well as "residents" need to deal with that uncertainty, too --> otherwise we'll have soon a big discussion about "increasing false alerts" and fatigue of the population. A general improvement on risk education incl. handling natural hazard event uncertainty is required.**

**Author Reply:** Thank you for this observation. Section 5.2 focuses on how uncertainty affected institutional interpretation and decision-making within Luxembourg's warning chain. We agree that uncertainty also influences how both officials and the wider public perceive and respond to warnings. While public reactions and alert fatigue were somewhat beyond the available evidence, we recognise that growing exposure to probabilistic forecasts and repeated warnings raises important communication challenges. In the revised discussion we will note that institutional treatment of uncertainty shapes public trust, and that improvements in risk education and communication are important to support both professional and citizen understanding of uncertainty in natural hazard events.

**List of relevant changes:**

**Lines 707-710:** 'How institutions handle uncertainty also shapes trust in warning systems. When uncertainty is communicated implicitly through procedural delay or conservative escalation, it may weaken confidence among both officials and the public. Repeated exposure to warnings that do not lead to visible action further raises communication and risk-education challenges, for decision-makers tasked with interpreting evolving signals under uncertainty.'
* * *
**RC1: If you intend to improve the reach and impact of your publication and would like to engage in a policy dialogue, it might be sensible to slightly change the title of the article. Currently you might be seen as implying that no action at all based on early warning was taken, whereas the reality in the "Bernd" event was more nuanced. Consider changing to "not enough protective" or "no impactful" action or the like.**

**Author Reply:** : Thank you for this helpful comment. The title "Signals Without Action" was chosen to capture the disconnect between available forecasts and the anticipatory measures that followed. Some action did occur, but not early or protective enough to prevent major impacts. The title refers to missed or delayed use of forecast information rather than a total absence of action. We appreciate the suggestion and will consider whether a minor adjustment or clarification in the abstract could better convey this nuance while preserving the central focus on the limited conversion of warning signals into timely protective measures.

**List of relevant changes:**

**Lines 13-14:** 'While response actions were taken during the event, they occurred too late or at insufficient scale to prevent major impacts.'
* * *
**RC1: Was it a disaster? Was there an official "disaster" declaration, or what makes you use the term disaster? Extreme event, yes, but disaster (also compared to Germany)?**

**Author Reply**: The term disaster is used intentionally. The July 2021 floods were formally declared a disaster by the Luxembourg government and met the UNDRR definition as a severe disruption exceeding local coping capacity. While no fatalities occurred and the scale was smaller than in neighbouring Germany for example, the impacts were unprecedented in Luxembourg's context, with extensive damage, evacuations, and emergency mobilisation across most of the country. Referring to it as an "event" would understate its national significance and overlook how disasters are defined relative to local capacity rather than absolute impact. We will clarify this distinction in the text to acknowledge both the difference in scale and the appropriateness of the term disaster for the Luxembourg case.

**List of relevant changes:**

**Lines 113-119:** 'In Luxembourg, the July 2021 floods were formally declared a 'natural disaster', reflecting the scale of impacts relative to national coping capacity rather than absolute losses. While the event was smaller in scale than the catastrophic flooding experienced in parts of Germany, it exceeded available response and recovery capacities in Luxembourg and constituted the most damaging flood event on record nationally.'
* * *
**RC1: Imprecise use of "hazard" - hazard is a general concept whereas what Early Warning should achieve is taking action towards an imminent event materializing**

**from that hazard, so correct the first occurrence of hazard and replace it with "(hazard) event" the 2nd time in that line**

**Author Reply:** Thank you for pointing this out. We agree and accept the distinction and will adjust the sentence to specify hazard event in the second occurrence.

**List of relevant changes:**

**Lines 27, 97, 642 :** Changed throughout whole manuscript (see tracked changes)
* * *
**RC1: Line 27: Name it - it is the "EW4ALL" Initiative**

**Author Reply:** We agree and will include "EW4ALL" .

**List of relevant changes:**

**Lines 29-30:** Recognising their significance, the United Nations has set an ambitious target through the Early Warnings for All (EW4All) initiative to ensure that by 2027, everyone on Earth should be covered by an Early Warning System (WMO, 2022).
* * *
**RC1: Line 33: Imprecise description of components for EWS - you might want to distinguish between EWS for climate-related hazards or hydromet hazards, since there are EWS for non-hydromet hazards as well.**

**Author Reply**: The description of Early Warning System components in this section refers specifically to hydrometeorological hazards, as this is the scope of the study. We agree that other types of EWS exist and will clarify this by specifying hydrometeorological where relevant.

**List of relevant changes:**

**Lines 31, 35, etc:** This has been corrected where applicable throughout the manuscript )see track changes).
* * *
**RC1: Line 36: A key point - you're missing the recipient of the message and the desired action they should take. The value chain should not just be observation-transmission-decision-making (seemingly by the authorities or Emergency Services) but start with the "first mile", the desired action taken by individuals, communities and society at large. EWS inherently should be a system comprising the human element and not just stay within a professional science-tech-public authority bubble. And I say this very consciously with a first-world, developed-context in mind - you are discussing "Bernd" in Luxembourg but the same thing just happened in Valencia, Spain as well so it is very much also a first world problem so we need to work on that last connection between officials taking decisions and the individual who needs to take action. Consider revising this point throughout the manuscript.**

**Author Reply:** We agree that this connection should be made more explicit. In the revised manuscript we will clarify in Section 2 that the purpose of the value-chain approach is to trace how forecast information moves through the system to its end users, and that the analysis of Luxembourg's 2021 floods shows how institutional design and communication processes

shaped whether information could reach and encourage action at the first mile. We will also note this more clearly in the discussion by highlighting that these structural conditions limited the translation of available warnings into better protective action among the population.

**List of relevant changes:**

**Lines 154-155:** 'It builds on the WMO WWR HIWeather Value Chain Project, which conceptualises Early Warning Systems as information chains that extend from forecast generation to community-level protective action, including measures taken by individuals, communities, and institutions.'

**Lines 552-555: '**In the national system, warnings are intended to reach residents through official dissemination channels, including public alerting systems, press communication, and institutional information platforms. This analysis focuses on the institutional conditions that shape whether public warnings can be authorised, issued, and disseminated, rather than on how residents interpret or respond to those warnings.**'**
* * *
> **RC1: Line 42, 46, 132 etc. etc: See general comment. If you introduce EWS as an acronym you can stay consistent. Why "Warning System" here, is this different from "early" warning system? Later, "early warning" but not system anymore.**

**Author Reply:** We agree that terminology consistency will improve readability. In the revision we will standardise wording to Early Warning System throughout, except where early warning clearly refers to the broader process rather than the system.

**List of relevant changes:**

**Lines 37, 45, 147, etc:** This has been changed to Early Warning System throughout except for when it describes the action of early warning.
* * *
> **RC1: Line 79: Word missing? Activating emergency - what? Protocols? Services?**

> **Author Reply:** Correct. The intended phrase was activating emergency plans and procedures. We will clarify this wording in the next revision.

**List of relevant changes:**

**Line 89:** emergency plans
* * *
> **RC1: Line 100: Between July 12 and July 15?**

**Author Reply:** Correct. The period referenced is between 12 and 15 July 2021, we will make the necessary changes.

**List of relevant changes:**

**Line 110:** Changes have been made see track changes.

**RC1: Line 104: Imprecise use of "frequent exposure" - don't understand the high time variability element of exposure. Exposure may be low or high and it may change over time, but it rarely fluctuates with a high frequency to make the term "frequent exposure" adequate?**

**Author Reply:** Agreed. The phrase frequent exposure was imprecise. The intended meaning was recurrent or regular exposure over time. We will adjust the wording accordingly in the next revision.

**List of relevant changes:**

**Lines 118-120** Luxembourg's position within a dense river network contributes to recurrent flood exposure, particularly in low-lying valleys and urbanised catchments.
* * *
**RC1: Line 110: Improve sentence. I am unclear what is meant by "higher levels" - warning levels based on thresholds? Or warnings did not reach higher levels of government in terms of the messages reaching further? If the former, you will need to introduce what the warning levels/thresholds in Luxembourg look like first before discussing what happened or did not with warning levels.**

**Author Reply:** The sentence refers to the official colour-coded alert levels rather than levels of government. We will clarify this by rephrasing the sentence to read: "Although forecasts were available, official warnings did not reach higher alert levels until shortly before impacts began to unfold.

**List of relevant changes:**

**Lines 125-126:** '...warnings did not reach higher colour-coded alert levels until shortly before impacts began to unfold.'
* * *
**RC1: Line 145: See general comment - I perceive the ultimate actor to be the individual, family or community to take action, not the government decision-maker to "tell them what to do" since often this involves still a technical message that does not lead to the desired action since it remains unclear what the desired action should be.**

**Author Reply:** We agree that the effectiveness of early warning also depends on individual and community action. The paper focuses on the institutional level because this is where decisions about warning content, format, and timing are made, which in turn determine how clearly the public can understand and act on them. We will clarify in the discussion that while the analysis centres on institutional processes, these directly influence whether people receive guidance that enables protective action.

**List of relevant changes:**

**Lines 553-555:** 'This analysis focuses on the institutional conditions that shape whether public warnings can be authorised, issued, and disseminated, rather than on how residents interpret or respond to those warnings.'

**Lines 718-723:** 'While effective early warning depends on whether warnings are understood and acted upon by residents, this analysis focuses on the institutional conditions that determine whether such warnings can be authorised, escalated, and disseminated in the first place.'

**Lines 748-750:** 'People-centred outcomes depend on how residents receive and act on warnings; the analysis focuses on the institutional processes that shape whether warnings can reach and support protective action, while resident-facing message design lies beyond the empirical scope.'
* * *
**RC1: Section 3.1: See general comment**

**Line 180: ", which" unnecessary.**

**Author Reply:** OK
* * *
**RC1: Line 189: In the interest of being more multi-hazard and clear on what's incl. and what's not: Can you specificy whether CGDIS is only responsible for such hazards as mentioned (severe weather, flooding) or wheter it would also respond in non-meteorological natural hazards situations?**

**Author Reply:** CGDIS is responsible for all types of emergencies and hazards in Luxembourg, including meteorological, hydrological and other natural hazard situations. We will clarify this in the text to specify that its mandate covers the full spectrum of civil protection hazards, not only severe weather and flooding.

**List of relevant changes:**

**Lines 205-207:** CGDIS operates within a multi-hazard civil protection framework, with responsibility for operational response to meteorological, hydrological,and other civil protection emergencies in Luxembourg.
* * *
**RC1: Line 193: Specify when it had not been implemented (during the July 2021 floods I assume) - or at the time of writing?**

**Author Reply:** The reference concerns the period of the July 2021 floods, when the measure in question had not yet been implemented.

**List of relevant changes:**

**Lines 210-211:** The PNOS was approved and signed in October 2021 and had not yet been implemented during the July 2021 flood event.
* * *
**RC1: Line 201: You contradict yourself - MeteoLux cannot issue alerts... only alerts from MeteoLux... ? I assume "only forecasts and warnings issued by MeteoLux are considered ..."?**

**Author Reply:** Meteolux publishes both weather warnings and flood warnings (the latter on behalf of the Water Management Administration), but its mandate is limited to the publication of these warnings based on defined physical thresholds. It does not have authority to activate emergency planning or response measures, which fall under the responsibility of the Haut-

Commissariat à la Protection Nationale (HCPN). The intended meaning of the sentence was that, under national law, only forecasts and warnings issued by MeteoLux are recognised as the official meteorological source for decision-making. We will rephrase this section to make the distinction between issuing warnings and activating emergency plans clearer.

**List of relevant changes:**

**Lines 220-222:** 'Meteorological forecasts and warnings issued by MeteoLux are recognised as the official basis for decision-making, while Crisis Unit activation are determined by the HCPN and the Prime Minister (HCPN Law, 2016; Ministry of State et al., 2015).'
* * *
**RC1: Table 2 - are these exact translations? I am surprised that for weather the terminology "danger" is used and for flood "risk". Especially the latter I find confusing since we are talking immediate, imminent river conditions leading to danger to lives and livelihoods, rather than the conceptual "risk".**

**Author Reply:** During the July 2021 floods, the meteorological and hydrological alert systems in Luxembourg used distinct terminologies. MeteoLux applied the descriptors Danger (orange) and Extreme Danger (red) for meteorological warnings, while the Water Management Administration used escalation phases for flooding: Pre-Alert Phase (Minor Flood Risk) orange, and Alert Phase (Major Flood Risk) red. The wording in Table 2 reproduces the official terminology in use at that time. However, the brackets around Pre-Alert Phase may have caused confusion. We agree that this should be clarified and will present it exactly as defined in the official framework. Minor and Major flood risk were the official descriptors of the corresponding alert levels. The flood terminology was updated nationally in late 2024 to mirror the meteorological colour-code categories, but these later changes are outside the scope of this study and not relevant to the situation in 2021. The terminology valid in July 2021 can be verified through the archived Infocrise website:

https://web.archive.org/web/20240930004300/https://infocrise.public.lu/en/inondations/phases-alerte.html

For all information about current governmental intervention plans please see www.infocrise.lu
* * *
**RC1: Line 279: July 17-18**

**Author Reply:** Ok
* * *
**RC1: Line 280: Side comment: I like the use of "insured damages" as according to the UNFCCC this would be the right, precise terminology. The industry would be using "insured losses" which according to UNFCCC definition makes no sense since they are reimbursed for recovery. We might want to influence the industry to adjust wording…**

**Author Reply:** Agreed. We will retain the term insured damages as suggested, consistent with UNFCCC definitions.
* * *
**RC1: Line 302: Correct, and remove track changes**

**Author Reply:** Ok
* * *
**RC1: Line 377: See general comment on people. This is the first time that you mention people ("residents") rather than stay with in the official chain, and it was not yet clarified how communication from EWS should lead to action by those residents. Consider revising with a people-centered re-focus.**

**Author Reply:** We will make this connection clearer by adding a short sentence where residents are first mentioned to explain how official warnings are intended to reach the public through established communication channels. In the discussion we will also note that the institutional processes analysed in this paper determine how effectively information can be translated into guidance that supports protective action by residents.

**This has been done. Line 552-555**
* * *
**RC1: Line 386: You mention France, but not in the outline of the study that France was one of the countries affected by Bernd.**

**Author Reply:** Agreed. We will note that France was also affected by the July 2021 floods to complete the regional context.

**List of relevant changes:**

**Line 111:** France has been added
* * *
**RC1: Paragraph line 468: I think you're missing a point here. Although the neighboring countries acted differently (and you recommend Lux. to do the same), it seemingly made no difference to how their EWS performed - those countries, particularly Germany, are the ones where so many lives were lost and people reported not received a (meaningful) warning…**

**Author Reply:** The comparison with neighbouring countries is meant to illustrate differences in institutional procedures, not performance outcomes. We will clarify that while Germany and Belgium experienced far greater impacts, their systems also faced serious challenges. The point is that procedural design influences how forecasts are handled, but this alone does not determine outcomes. We will adjust the phrasing to make this distinction clear.

**List of relevant changes:**

**Lines 497-499:** 'In neighbouring countries, Early Warning Systems operated under different procedural criteria, allowing alert decisions to draw on exceedance across regional observation networks and convergence within ensemble forecast products, leading to earlier issuance of red-level alerts on 13 July.'
* * *
**RC1: Line 521: I feel this is a bit jumping to conclusions - the public needs actionable messages, not just an alert. You may want to go back to the discussion on Gouv-Alert earlier and review not only if/when it was activated or not (the tech glitch), but also what it would have done without it - would these have been actionable messages or still unclear what action to take? Compare with your reference to Spain's DANA, where the system technically worked, but was both activated late AND did not incl. clear messages tailored to different audiences.**

**Author Reply:** We will revisit the section on GouvAlert to clarify that the issue was not only its non-activation but also the nature of the messages it delivers. We will also reference the Spain DANA example to underline that technical activation alone does not ensure effective communication or behavioural response.

**List of relevant changes:**

**Lines 126-128:** 'Challenges in communication, including a warning notification via the GouvAlert mobile system that was not delivered and delays in institutional coordination, contributed to ambiguity regarding responsibilities and the actions expected of different actors.'

**Lines 516-517:** During the July 2021 flood, Luxembourg's public alerting systems were not used in a way that enabled timely protective action.

**Lines 543-544:** The 2024 DANA floods in Valencia illustrate how procedural communication choices, including alert timing and message content, can limit the protective value of public warnings
* * *
> **RC1: Line 547 and 550, but also refer to comment on choice of the title - if you want to get policy action you may want to tone done - policy makers in Lux may justifiably say that you suggest Germany did better but actually despite having the advantage in 2021 that you suggest Lux also improves on the result in Germany was not better at all, so this is not the decisive factor...**

**Author Reply:** The intention is not to suggest that Germany performed better, but to highlight differences in system design and mandate structure rather than outcomes. We will adjust the phrasing in this section to ensure that the comparison is presented neutrally and does not imply relative performance, but instead focuses on how procedural design shaped the handling of forecasts in each country.

**List of relevant changes:**

**Lines 581-582:** 'These signals, documented in post-event evaluations in Germany and Belgium,...'

**Lines 584- 591:** Although EFAS and EFI were monitored internally by AGE and MeteoLux, no procedural framework existed in Luxembourg for using these products to inform warning level decisions and public communication. In Germany and Belgium, post-event analyses describe procedural arrangements that allowed ensemble-based and regional information to be considered within warning processes, without implying more effective outcomes. In Luxembourg, the absence of an equivalent framework meant that these forecasts remained outside formal decision pathways, and no institutional review has clarified how such inputs could be interpreted or integrated.
* * *
> **RC1: Line 567: Suggest you revise to the 5 steps of the disaster risk management (DRM) cycle. Also, don't use "mitigation" in this context but (corrective and prospective) risk reduction. Mitigation in today's climate crisis should be reserved as a term for climate mitigation (CO2) rather than use it confusingly as a seeming synonym for risk reduction.**

**Author Reply:** We agree to replace mitigation with risk reduction where appropriate. However, we do not adopt the cyclical DRM model, as the paper conceptualises disasters as systemic

outcomes of governance and design rather than sequential phases. This approach aligns with (Bosher et al 2021) DOI: https://doi.org/10.1108/DPM-03-2021-0071

**List of relevant changes:**

**Lines 607, 646:**  Mitigation has been replaces with risk reduction where appropriate
* * *
**RC1: Line 580: I assume "response" here is not meant sensu strictu (responding to crisis) but rather as a response to the trigger -therefore I would use "action" in line with your title etc.**

**Author Reply**: Ok, We will adjust the wording to action for consistency with the paper's framing and title.

**List of relevant changes:**

**Line 607-608:**
* * *
**RC1: Line 592: Not hazard but (hazard) event**

**Author Reply:**  Agreed. We will replace hazard with hazard event at this point to reflect the more precise terminology.

This has been done (**Lines 27, 97, 642)**
* * *
**RC1: End of 6.3, and conclusion in 7: Nothing to challenge the conclusion as stated per se, but I am lacking again the final step from officials being "warned" or informed to reaching the "resident" that you prominently mention elsewhere - you could outline that maybe this was outside of the scope of your work and the suggested model, but that more is needed on designing and reviewing how targeted messages must be developed/improved to really get protective action by those "residents".**

**Author Reply:**  Thank you and we agree. In the revision we will clarify at the end of Section 6.3 and in the conclusion that the study focuses on institutional processes up to the point where information is issued to the public. We will note that the next step, designing and evaluating how messages reach and support residents in taking protective action lies beyond the scope of this analysis but represents an essential direction for further research and system improvement.

**List of relevant changes:**

**Lines 718-723:** 'While effective early warning depends on whether warnings are understood and acted upon by residents, this analysis focuses on the institutional conditions that determine whether such warnings can be authorised, escalated, and disseminated in the first place. The design, targeting, and evaluation of resident-facing messages are therefore recognised as essential, but lie beyond the empirical scope of this study. A more detailed mapping of domain-specific processes and interpretive practices within institutions would require data beyond those available for this analysis and represents a priority direction for future research.'

**Lines 748-754:** 'People-centred outcomes depend on how residents receive and act on warnings; the analysis focuses on the institutional processes that shape whether warnings can

reach and support protective action, while resident-facing message design lies beyond the empirical scope.'

**CC1 COMMUNITY REPLY INCLUDING LIST OF RELEVANT CHANGES**

**CC1: This is a very well written paper on an important case study of a small country, Luxembourg, affected by the 2021 floods in Western Europe. The paper highlights important issues of coordination, as well as warning and governance failures that are highly interesting also to an international audience. While the paper does a good job of documentation and going in detail by providing a great overview over planning documents and detailing flood and warning aspects, it also has certain shortcomings which I would like to address to suggest improvements for the paper.**

**More explanation and maybe even visualization is recommended for the value chain analysis and model. This seems to be the conceptual background and backbone selected for the paper and in some descriptive way it is taken up again here and there. But a consistent usage, and maybe more explicit visualization or string and use of certain components could be useful to even better align the story of the paper. The two sources provided also did not help me to understand the value chain model approach, so please add more sources.**

**As a novelty and interesting name, the paper suggests a water drop model. It does not become fully clear whether this model has been independently developed by the group of researchers, or whether it is a model already developed, and for whom and for which purposes. Reading the paper, it mainly seems to be represented by an illustrated graphic. The graphic is nice, but appears to be mainly a combination of a system representation with a boundary and input and output features. It also contains part of a subsystem represented as a triangle in which a tree model like an event three leads to a single output that is then connecting to the warning. It is especially this vector connecting to warning that is emphasized as the main failure in the whole impact chain. While this is plausible, one might also argue that the usage of representation of the tree model makes it look like this was an inevitable outcome. This is a schematic representation problem of similar diagrams used in quality management, such as the Ishikawa diagram or tree models or bowtie analysis and others. Critic around such pipe, or tunnel approaches, or "Nürnberger Trichter" might not be known to a larger international audience and I just want to leave this as a hint for the writers of the study. I think no defense is needed here and it does not need to be modified in the paper. But I must confess that I don't think that the diagram is so very novel as a method and I think it is at least in some parts suggestive, but which is OK. however, to call this a water drop model might be a little bit overstretched in my opinion.**

**What is more important is to ask which of the contributing factors on the left side were key in the final outcome that the warning failed? And if and how they can be compared to each other? The description shows this, but I'm not sure that the part with the threshold is really so convincing. Is this really a threshold problem or not just also a problem of observation or governance and therefore already covered by the two other main contributing factors that led to the failure?**

**I think it has more potential because also in other areas in Germany, the misinterpretation of such rain and river gauges, and prediction curves were part of the problem. Maybe it could be categorized into certain domains of problem types**

**to make the three contributing factors a bit easier to differentiate. The threshold problem could be a communication, interpretation, or transfer of knowledge problem maybe?**

**I would also suggest to make more usage of the very nice and inspiring diagram from Brian Golding here. Which major gaps of knowledge domains and missing bridges between them are the main aspect behind the three contributing factors analyzed? And again, which types of values and which type types of chains or components of a chain are analyzed here and also in your water drop model diagram? I think there's more in that study and idea than what is already described, which is why I encouraged to think about it once again.**

**Finally, the paper is mainly descriptive like an extended report version. This is also very good and necessary since publications about Luxembourg and the flood damages in 2021 are scarce and the paper therefore is a very important contribution. The writers also made a very good job, but I would encourage to include some more features of scientific assessment such as adding a limitation section to the discussion. It would be very illustrative for other readers to understand how the approach of the value chain, and the application of the water drop model have also shown certain challenges, which might help others who use the approach to better conduct it themselves.**

**Maybe also some overarching research question or guiding thoughts from the beginning of the paper could be taken up more prominently at the end or of a summary of main innovations or insights could be given.**

**My apologies for extended comments, but I hope it does not prompt for large additional sections. I'm looking forward to the publication of this paper.**

**Author Reply:** We thank the commenter sincerely for taking the time to read our paper in such detail and for offering thoughtful, constructive feedback. We are very grateful for the encouraging words about the paper's contribution and for the care with which conceptual aspects were considered. The suggestions are valuable and help us reflect further on how to make our analytical approach clearer and more accessible to readers.

The value-chain framework forms the conceptual backbone of the paper. Following the WMO/WWRP Value Chain Project (WMO, 2024; Ebert et al., 2023; Hoffmann et al., 2023), we understand early warning as a process linking forecast generation, translation, communication, and decision. This framing structures both the empirical reconstruction (Sections 2–4) and the analytical discussion (Sections 5–6). We applied the framework operationally, using its nodes–actors–flows logic and the Database Questionnaire developed under the WMO/WWRP HIWeather project (Hoffmann et al., 2023) to guide data collection and the organisation of results. This questionnaire, designed to document the end-to-end flow of information and decision-making in high-impact weather events, informed how we traced actors, information products, decisions, and outcomes across Luxembourg's warning system. It enabled us to identify where forecast value was maintained or constrained along the chain. The official WMO Value-Chain Framework and supporting materials are publicly available via the WMO Library (https://library.wmo.int/idurl/4/69103)

We developed the Waterdrop Model for this study. It extends the reconstructed value chain and translates its logic into a diagnostic structure that helps explain why forecast information was not converted into better anticipatory protective action. Its purpose is to visualise how institutional design and procedural thresholds influence the operational use of information. In doing so, the model aligns with calls to move beyond linear or cyclical representations of disaster processes (Neal, 1997; Bosher et al., 2021). The Waterdrop Model conceptualises disaster risk as a multi-layered process shaped by governance and mandate boundaries, not by event sequences alone.

More information here:

Bosher, L., Chmutina, K., & van Niekerk, D. (2021). Stop going around in circles: towards a reconceptualisation of disaster risk management phases. Disaster Prevention and Management: An International Journal, 30(4/5), 525-537.

Neal, D. M. (1997). Reconsidering the phases of disaster. International Journal of Mass Emergencies & Disasters, 15(2), 239-264.

We appreciate the insightful remarks regarding the potential for schematic forms to appear deterministic. This observation is well taken. The comparison with Ishikawa and bow-tie diagrams is helpful, as it underlines the importance of clarifying that the model shows interacting processes, not deterministic cause-effect chains. *We will refine the caption and accompanying text accordingly to make this interpretive purpose more clear*.

**List of Changes:**

**Lines 616-618:** 'The model was developed as a diagnostic extension of the reconstructed value chain and is intended to examine how institutional design conditions the use of forecast information and is not intended as a prescriptive or deterministic framework.'

**Lines 642-643:** 'This figure provides a conceptual representation of how warning systems process and filter forecast information.'

We value the question about the relative weight of the contributing factors. The "threshold" aspect is not purely technical but procedural, referring to institutional points at which interpretation and responsibility shift. *We will clarify in Section 6 that thresholds intersect with governance and interpretation and that their role is systemic, not isolated*.

**List of Changes:**

**Lines 143, 228, 248, 254, etc:** Procedural thresholds

**Lines 633-634:** 'These thresholds function as institutional decision rules that intersect with governance arrangements by defining when responsibility shifts from monitoring to authorisation and action.'

The suggestion to link the analysis more closely to Golding et al.'s "valleys of death" diagram is appreciated and well taken. The WMO/WWRP Value Chain framework applied in this paper was itself influenced by Golding et al.'s work, which conceptualises the gaps between scientific knowledge, service delivery, and decision-making. The Waterdrop Model builds on this foundation by illustrating how such gaps manifested in Luxembourg's warning system through institutional mandates and communication boundaries. *We will make this conceptual link explicit in the revised text*. A detailed mapping of individual knowledge domains, as proposed in

the comment, would require additional data and targeted analysis beyond the scope of the present study, but it represents a promising direction for future research.

**List of Changes:**

**Lines 619-618:** 'The model was developed as a diagnostic extension of the reconstructed value chain and is intended to examine how institutional design conditions the use of forecast information and is not intended as a prescriptive or deterministic framework.'

We emphasise that although the paper contains detailed documentation, its aim is analytical. The reconstruction of the 2021 floods provides the empirical basis to explain how institutional structure and mandate configurations shaped forecast interpretation. The integration of the Value Chain approach and the Waterdrop Model provides a systematic means to analyse these mechanisms. *We will include a short paragraph on methodological limitations in the discussion to outline the scope of available evidence and the interpretive boundaries of the Waterdrop Model.*

**List of relevant changes:**

**Lines 718-723:** 'While effective early warning depends on whether warnings are understood and acted upon by residents, this analysis focuses on the institutional conditions that determine whether such warnings can be authorised, escalated, and disseminated in the first place. The design, targeting, and evaluation of resident-facing messages are therefore recognised as essential, but lie beyond the empirical scope of this study. A more detailed mapping of domain-specific processes and interpretive practices within institutions would require data beyond those available for this analysis and represents a priority direction for future research.'

**Lines 728-735:** 'This analysis is based on publicly available records, institutional documentation, and reconstructed timelines, and is therefore limited to formally documented procedures, mandates, and authorised communication channels within the national warning system. Informal decision-making, undocumented interpretations, and internal deliberations are not captured. The analysis further focuses on the warning system up to the point at which alerts are issued to the public. Public interpretation, behavioural response, and message effectiveness are not examined, as these dimensions require different data and methods. These limitations should be considered when interpreting the findings. They also highlight an important direction for future research on people-centred early warning, in which institutional analysis is complemented by studies of public understanding and response.'

We appreciate the suggestion to restate the overarching research question more prominently at the end. In the revised conclusion, *we will explicitly return to the central question of how forecast signals were or were not translated into better anticipatory action and summarise that institutional design largely determined the space for action*. This synthesis will highlight the broader relevance of combining the Value Chain framework with the Waterdrop Model to analyse governance in early-warning systems.

**List of Changes:**

**Lines 752-756:** 'The analysis returns to the central question of how forecast signals were translated into anticipatory action during the July 2021 floods in Luxembourg. The findings show that institutional design largely determined whether early information could be authorised, interpreted, and acted upon in time. The Value Chain approach and the Waterdrop Model show

how governance structures shape the operational value of forecasts in Early Warning Systems across different institutional settings.'

We thank the commenter once again for the generous engagement and constructive feedback. The comments have been read with care and will directly guide improvements to the paper. We are grateful for the opportunity to clarify these points and for the collegial spirit in which the feedback was offered.
* * *
**RC2 REVIEWER REPLY INCLUDING LIST OF RELEVANT CHANGES**

**RC2: The authors present a detailed analysis of the response of the national early warning system to the 2021 floods in Luxembourg. The analysis identifies a number of issues in the official procedures that led to a delayed emergency response. The authors also introduce a conceptual model aimed at identifying possible problems in the monitoring, warning and emergency management chain, and applied it to the case study of 2021 floods.**

**Overall this is an interesting work and I enjoyed reading it. Having read the comments already posted by Referees #1 and #2, I'll focus my comments on those parts that have not been addressed.**

**Author Reply:** We thank the reviewer for reading the manuscript and for the positive assessment of the study. We appreciate the reviewer's focus on aspects not covered by the other referees. We respond to each point raised below and indicate how the manuscript will be revised.

**RC2: Major points**

**RC2: Description of the Water drop model: my impression is that the added value of the proposed conceptual model does not emerge clearly from the description. Presently, the conceptual model seem to be focused on the uptake (or not) of incoming forecasts, due to the filtering of information in the monitoring and emergency management chain.**

**Author Reply:** We agree with the reviewer that the added value of the Waterdrop Model is not yet stated clearly enough in the manuscript. In the revised manuscript, we will clarify that the purpose of the Waterdrop Model is to explain how institutional design, including official data sources, procedural thresholds, and mandated responsibilities, shapes whether available signals can become actionable. The model complements the value chain analysis by focusing on how institutional and procedural constraints limit the use of available forecast signals, rather than on forecast interpretation itself.

**List of relevant changes:**

**Lines 616-618:** 'The model was developed as a diagnostic extension of the reconstructed value chain and is intended to examine how institutional design conditions the use of forecast information and is not intended as a prescriptive or deterministic framework.'

**Lines 633-634:** 'These thresholds function as institutional decision rules that intersect with governance arrangements by defining when responsibility shifts from monitoring to authorisation and action.'

**Lines 748—749:** 'The Waterdrop Model explains how mandates and responsibilities shape whether forecast information can lead to action.'

> **RC2: While this is indeed an important topic, other important aspects are loosely represented, such as the timing for warning dissemination, the linkage between the observed timeline of the event and the actions taken, as well as other constraints that might act as bottlenecks in the procedure.**

**Author Reply:** We agree and we will revise the description of the Waterdrop Model to incorporate these elements more explicitly, clarifying how timing and procedural constraints shape whether forecast information leads to action within the warning system. This will strengthen the conceptual alignment between the empirical reconstruction of the 2021 event and the model's analytical purpose.

**List of relevant changes:**

**Lines 636-637:** 'Within this structure, procedural bottlenecks can delay escalation and dissemination even when risk information is available and technically credible.'

**Lines 644-646:** 'As time passes and certainty increases, more signals may enter the triangle, but the decision space for anticipatory action narrows, increasing the risk that warnings are issued once impacts are already unfolding.'

> **RC2: For instance, the authors report that the precipitation measured at Findel meteo station was the highest ever recorded, yet it did not result in a warning procedure. This seems to indicate that the threshold was not adequately set, and such potential vulnerability of the system should also have a place in the conceptual model (i.e. a system that rely on one single station should emerge as more vulnerable from the conceptual model).**

**Author Reply:** We agree with the reviewer that the implications of relying on a single reference station and fixed thresholds are not yet sufficiently explicit in the presentation of the Waterdrop Model. This issue is analysed in Sections 4 and 5, its role as a structural vulnerability could be explained better in the discussion. In Section 6.1, we will clarify that the Waterdrop Model reflects how the definition of valid data sources and thresholds shapes what information can trigger action and that systems centred on a single reference station are more vulnerable to threshold design. In Section 6.2, we will link the Findel station example to the model by explaining how record precipitation could remain operationally ineffective because it did not trigger a change in warning level under the existing procedure. We will also adjust the caption and accompanying text of Figure 8 to make clear that restricting the monitoring basis to a single station represents a narrowing of the decision space, which increases the risk of delayed response.

**List of relevant changes:**

**Lines 650-651:** 'In Luxembourg, this structural filtering was reinforced by reliance on a single reference station at Findel, which concentrated procedural authority in one location and increased vulnerability to delayed threshold exceedance'

**Lines 672-674:** 'Despite record precipitation at Findel and extreme rainfall recorded at other stations, no procedural mechanism existed to escalate warnings based on broader observational or probabilistic evidence.'

> **RC2: As another example, the format of the timeline adopted in Figure 8 might be fine for the conceptual scheme, but I find it confusing when applied to the case study of Luxembourg floods. Perhaps a standard timeline similar to Figure 7 could be more effective to show the effectiveness of system in reacting to ongoing situation and providing timely response.**

**Author Reply**: We acknowledge that the distinction between Figures 7 and 8 was not sufficiently clear. In response, we will add explicit explanatory text to guide the reader, clarifying the different purposes of the two figures and how they should be read and cross-referenced. Figure 7 will be clearly framed as the empirical event timeline, while Figure 8 will be more explicitly positioned as a conceptual representation of the warning system.

**List of relevant changes:**

**Lines 625-627:** 'Figures 7 and 8 are intended to be read together, with Figure 7 documenting the empirical sequence of forecasts and alerts during the event and Figure 8 providing a conceptual framework for interpreting how the warning system processed that information.'
* * *
> **RC2: To summarize, I invite the authors to expand the scope and the description of the conceptual model.**

**Author Reply**: We agree with the reviewer that the scope and description of the conceptual model require some clarification. In the revised manuscript, we will expand its purpose and how it complements the reconstruction. This will include clearer explanation of how procedural dependencies are reflected in the model, and how these elements help explain the observed response during the July 2021 flood.

**List of relevant changes:**

**Lines 708-711:** 'How institutions handle uncertainty also shapes trust in warning systems. When uncertainty is communicated implicitly through procedural delay or conservative escalation, it may weaken confidence among both officials and the public. Repeated exposure to warnings that do not lead to visible action further raises communication and risk-education challenges, for decision-makers tasked with interpreting evolving signals under uncertainty.'

**Lines 718-723:** 'While effective early warning depends on whether warnings are understood and acted upon by residents, this analysis focuses on the institutional conditions that determine whether such warnings can be authorised, escalated, and disseminated in the first place. The design, targeting, and evaluation of resident-facing messages are therefore recognised as essential, but lie beyond the empirical scope of this study. A more detailed mapping of domain-specific processes and interpretive practices within institutions would require data beyond those available for this analysis and represents a priority direction for future research.'
* * *
> **RC2: Section 4.2: this section broadly describes the effects of meteorological conditions. However, the exact timeline of flooding processes is not well explained.**

**It is said that "The SPC issued a yellow vigilance alert at 14:30 on 13 July, upgraded to orange by midday on 14 July and to red at 17:15 the same day", but it is is not clear how these actions related with the actual situation on the ground; for instance, when did flooding actually begin? When was the peak flow observed in the affected rivers?**

**Author Reply:** We note that Section 4.2 already describes the onset and evolution of flooding, including the timing of rising water levels and prolonged peak conditions in several catchments. However, we agree that the relationship between these hydrological developments and the timing of SPC alert changes is not made sufficiently explicit. In the revised manuscript, we will clarify this linkage by more directly aligning the alert chronology with the observed onset and peak of flooding.

**List of relevant changes:**

**Lines 291-295:** 'At the time of the yellow level alert on 13 July, river levels were already increasing across several catchments. Flooding began during the early hours of 14 July as rainfall intensified and runoff accumulated. By the time the red level alert was issued in the late afternoon of 14 July, flooding was already affecting multiple river systems, with water levels continuing to rise and peak conditions extending into 15 July.'
* * *
**RC2: Minor points**

**RC2: Figure 4: there are two different green markers and two turquoise lines, can you better explain what they represent?**

**Author Reply:** In Figure 4, the green symbols represent observational information. The green hourglass symbol shows the mean of the station observations within the 1° box, while the green dot represents the proxy analysis used to complement the observational record. The turquoise horizontal lines indicate fixed reference levels derived from the model climate and are used as benchmarks for comparing observed and forecast precipitation amounts. We will revise the figure caption to make the meaning of these elements explicit.

**List of Changes:**

**List of relevant changes:**

**Figure 4 caption Line 279:** 'Blue box-and-whisker plots represent the distribution of IFS ensemble forecast members (IFS-ENS) for each forecast date. Red dots indicate the deterministic control forecast, and black triangles show the maximum value of the IFS model climate (M-climate). Cyan box-and-whisker plots represent the IFS model climate for the corresponding period. Green hourglass symbols represent the mean of station observations within the 1° box, while green dots represent a proxy precipitation analysis used where direct observations are spatially or temporally limited. The two turquoise horizontal lines correspond to fixed reference thresholds derived from the model climate.'
* * *
**RC2: Figure 7: The timeline currently only reports the evolution of available forecasts and alerts during the event. Adding in the timeline a description of observed weather conditions and consequent impacts (flooding conditions, evacuations and other responses to the emergency) would help in better assesssing the performance of the national alert system.**

**Author Reply:** We agree that adding observed impacts could provide additional context, but this is outside the intended scope of Figure 7. Figure 7 documents the official timeline of forecasts and alert levels used for the forensic reconstruction of the warnings, while observed conditions and impacts are described elsewhere in the manuscript. Figure 8 is used to interpret how the system operated as the event unfolded. We will clarify these distinct roles in the text and figure captions.

**List of relevant changes:**

**Figure 7 caption Line 349:** 'This figure presents the empirical timeline of available forecasts and officially issued alerts during the event. Alerts are shown for MeteoLux (weather) and AGE (flood) with triangle markers indicating forecast issuance time and coloured bars representing alert validity periods. Observed impacts and response actions  are not represented in this figure. This figure chronology is based on official bulletins and institutional records (AGE, 2021a; MeteoLux, 2021a; Gouvernement du Grand-Duché de Luxembourg, 2023).'

**Lines 625-627:** 'Figures 7 and 8 are intended to be read together, with Figure 7 documenting the empirical sequence of forecasts and alerts during the event and Figure 8 providing a conceptual framework for interpreting how the warning system processed that information.'

**Figure 8 caption Line 642:** 'This figure provides a conceptual representation of how warning systems process and filter forecast information.'
* * *
**RC2: Line 676, Conclusions: consider replacing "as we demonstrate" with " as demonstrated by the severe floods of July 2021 in Luxemboug,…"**

**Author Reply:**  We will revise the wording in the conclusion as suggested.

**List of relevant changes:**

**Lines 734-735:** 'As  demonstrated by severe floods of July 2021 in Luxembourg, having forecast information available does not guarantee that early action will follow.'